# Rethinking Semi-Supervised Imbalanced Node Classification from Bias-Variance Decomposition

**Divin Yan**[†], **Gengchen Wei**[†], **Chen Yang**[†], **Shengzhong Zhang**[†], **Zengfeng Huang**[†*]

[†]Fudan University, {yanl21, gcwei22, yanc22}@m.fudan.edu.cn
{szzhang17, huangzf}@fudan.edu.cn

## Abstract

This paper introduces a new approach to address the issue of class imbalance in graph neural networks (GNNs) for learning on graph-structured data. Our approach integrates imbalanced node classification and Bias-Variance Decomposition, establishing a theoretical framework that closely relates data imbalance to model variance. We also leverage graph augmentation technique to estimate the variance, and design a regularization term to alleviate the impact of imbalance. Exhaustive tests are conducted on multiple benchmarks, including naturally imbalanced datasets and public-split class-imbalanced datasets, demonstrating that our approach outperforms state-of-the-art methods in various imbalanced scenarios. This work provides a novel theoretical perspective for addressing the problem of imbalanced node classification in GNNs.

## 1 Introduction

Graphs are ubiquitous in the real world, encompassing social networks, financial networks, chemical molecules [22, 46, 9] and so on. Recently, graph neural networks (GNNs) [16, 35, 9] have shown exceptional proficiency in representation learning on graphs, facilitating their broad application in various fields. Nevertheless, the process of learning on graphs, analogous to its counterparts in computer vision and natural language processing, frequently encounters significant obstacles in the form of skewed or insufficient data, leading to the prevalent issue of class imbalance. It is undeniable that real-world data often exhibits biases and numerous categories, which entail data limitations and distribution shifts to be common. Furthermore, coupled with the fact that GNNs often feature shallow architectures to prevent "over-smoothing" or "information loss", training with such datasets leads to severe over-fitting problems in minor classes.

Designing GNN imbalanced learning schemes for graph-structured data poses unique challenges, unlike image data. Graph-structured data often requires consideration of the data's topology and environment, and empirical evidence [6, 24, 31] shows that topological asymmetries can also affect the model's performance. However, due to the extremely irregular and structured nature of the data, quantifying and solving topological asymmetry is difficult and computationally intensive. As a result, existing methods such as oversampling [28, 50, 24] and loss function engineering [6, 31] are problematic for achieving satisfactory outcomes. In depth, the modeling approach for data personalization exhibits very poor scalability and generalization capability. Furthermore, the prevalence of this problem has prompted the community to seek a more effective framework for imbalanced learning. ***Thus, a more fundamental and theoretical perspective is urgently needed when considering the imbalance node classification problem.***

---

[*]Corresponding Author.

37th Conference on Neural Information Processing Systems (NeurIPS 2023).

Following the idea, in this work, we propose a novel viewpoint to understand graph imbalance through the lens of *Bias-Variance Decomposition*. The Bias-Variance Decomposition [1, 2, 23, 45] has long been applied in terms of model complexity and fitting capacity. Our theoretical analyses have confirmed the relationship between model variance and the degree of dataset imbalance, whereby an imbalanced training set can lead to poor prediction accuracy due to the resulting increase in variance. Furthermore, we conducted some experiments on real-world dataset and the results demonstrate a significant relationship between the variance and the imbalance ratio of the dataset. As far as we know, we were the first to establish a connection between imbalanced node classification and model variance, which conducts theoretical analysis in this field.

Moreover, we have devised a regularization term for approximating the variance of model, drawing on our theoretical analysis. The main challenge of this idea lies in estimating the expectation across different training sets. Our key insight is to leverage graph data augmentation to model the different training sets in the distribution, which has not been explored before. Our empirical evaluations have consistently demonstrated that our algorithm, leveraging three basic GNN models [16, 35, 9], yields markedly superior performance in diverse imbalanced data settings, surpassing the current state-of-the-art methods by a substantial margin. Notably, we have conducted meticulous experiments on two naturally imbalanced datasets, which serve as the realistic and representative benchmarks for real-world scenarios.

Our contribution can be succinctly summarized as follows: **(i)** We are the first to integrate imbalanced node classification and *Bias-Variance Decomposition*. Our work establishes the close relationship between data imbalance and model variance, based on theoretical analysis. **(ii)** Moreover, we are the first to conduct a detailed theoretical analysis for imbalanced node classification domain. **(iii)** Our principal insight is to leverage graph data augmentation as a means of representing the varied training sets that lie within the distribution, whilst simultaneously designing a regularization term that approximates the model's variance. **(iv)** We conduct exhaustive tests on multiple benchmarks, such as the naturally imbalanced datasets and the public-split class-imbalanced datasets. The numerical results demonstrate that our model consistently outperforms state-of-the-art machine learning methods and significantly outperforms them in heavily-imbalanced scenarios.

## 2 Preliminaries

### 2.1 Notations

We concentrates on the task of semi-supervised imbalanced node classification within an undirected and unweighted graph, denoted as $\mathbf{G} = (\mathbf{V}, \mathbf{E}, \mathbf{V_L})$. $\mathbf{V}$ represents the node set, $\mathbf{E}$ stands for the edge set, and $\mathbf{V_L} \subset \mathbf{V}$ denotes the set of labeled nodes. The set of unlabeled nodes is denoted as $\mathbf{V_U} := \mathbf{V} \setminus \mathbf{V_L}$. The feature matrix is $\mathbf{X} \in \mathbb{R}^{n \times f}$, where $n$ is the number of nodes and $f$ is the feature dimension. The adjacency matrix is denoted by $\mathbf{A} \in \{0, 1\}^{n \times n}$. Let $\mathbf{N}(v)$ represent set of adjacent nodes to node $v$. The labeled sets for each class are denoted as $(\mathbf{C}_1, \mathbf{C}_2, \dots, \mathbf{C}_k)$, where $\mathbf{C}_i$ represents the labeled set for class $i$. The imbalance ratio $\rho$, is defined as $\rho = \max_i |\mathbf{C}_i| / \min_i |\mathbf{C}_i|$.

### 2.2 Graph Dataset Augmentations

To craft diversified perspectives of the graph dataset, we deploy advanced graph augmentation methodologies, leading to the formation of $\tilde{\mathbf{G}} = (\tilde{\mathbf{A}}, \tilde{\mathbf{X}})$ and $\tilde{\mathbf{G}}' = (\tilde{\mathbf{A}}', \tilde{\mathbf{X}}')$. Elements such as node attributes and connections within the foundational graph are selectively obfuscated. These augmented perspectives are subsequently channeled through a common Graph Neural Network (GNN) encoder, symbolized as $f_\theta : \mathbb{R}^{n \times n} \times \mathbb{R}^{n \times f} \to \mathbb{R}^{n \times d}$, in order to distill compact node-centric embeddings: $f_\theta(\tilde{\mathbf{A}}, \tilde{\mathbf{X}}) = \mathrm{h} \in \mathbb{R}^{n \times d}$ and $f_\theta(\tilde{\mathbf{A}}', \tilde{\mathbf{X}}') = \mathrm{h}' \in \mathbb{R}^{n \times d}$.

### 2.3 Related Work

**Semi-Supervised Imbalanced Node Classification.** Numerous innovative approaches[28, 40, 50, 19, 25, 6, 24, 31, 43] have been proposed to tackle the difficulties arising from imbalanced node classification on graph data. GraphSMOTE [50] employs the SMOTE [5] algorithm to perform interpolation at the low dimensional embedding space to synthesize the minority nodes, while ImGAGN [25] introduces GAN [8] for the same purpose. Another work GraphENS [24] synthesizes

mixed nodes by combining the ego network of minor nodes with target nodes. To address topology imbalance in imbalanced node classification task, ReNode [6] adjusts node weights based on their proximity to class boundaries. TAM [31] leverages local topology that automatically adapts the margins of minor nodes. Due to spatial constraints, we provide a comprehensive exposition of other relevant literature in Appendix A.

## 3 Theory

### 3.1 Theoretical Motivation

This section presents a succinct overview of the classical Bias-variance Decomposition [1, 2, 23, 45] and its application to semi-supervised node classification. Furthermore, we illustrate the influence of imbalance on classification results, which subsequently amplifies the variance.

**Bias-variance Decomposition.** Let $x \in X$ and $y \in Y$ denote the input and label, respectively. Consider an underlying mapping $f : X \to Y$ and the label can be expressed as $y = f(x) + \epsilon$, where $\epsilon$ is some noise. Given a training set $D = (x_i, y_i)$, we train a model $\hat{f}(x) = \hat{f}(x; D)$ using the samples in $D$.

**Definition 1** *Bias-Variance Decomposition is that the expected predicted error of $f$ can be decomposed into Eq(1),*

$$\mathbb{E}_{D,x}\left[(y - \hat{f}(x; D))^2\right] = \left(\text{Bias}[\hat{f}(x)]\right)^2 + \text{Var}_D[\hat{f}(x; D)] + \text{Irreducible error}, \quad (1)$$

*where* $\text{Bias}[\hat{f}(x)] = \mathbb{E}_x[\hat{f}(x; D) - f(x)] = \mathbb{E}_D[\hat{f}(x; D)] - \text{E}[y(x)]$, *and* $\text{Var}_D[\hat{f}(x; D)] = \mathbb{E}_D\left[\left(\mathbb{E}_D[\hat{f}(x; D)] - \hat{f}(x; D)\right)^2\right]$.

It is important to note that this expectation is taken with respect to both the training sets $D$ and the predicted data $x$. The bias term reflects the model's ability to fit the given data, while the variance term indicates the stability of the model's results with different training sets. In other words, the variance describes the model's generalization performance.

Next, we consider the specific formulation of variance in the setting of semi-supervised node classification on graph. First, we make some assumptions to simplify the analysis.

**Assumptions.** We make two assumptions in our approach. Firstly, we assume that the node embeddings $h^i$ of node $x^i$ extracted by a graph neural network for nodes belonging to class $i$ follow a multivariate normal distribution $h^i \sim N(\mu^i, \Lambda^i)$, where $\Lambda^i$ is a diagonal matrix for all $i = 1, 2, \ldots, c$. Here, let $\epsilon^i = h^i - \mu^i$ which follows the distribution $\epsilon^i \sim N(0, \Lambda^i)$.

Secondly, we consider a simple classifier that estimates the probability of a node belonging to a particular class based the distance $h^T C^i$. Here, $C^i = C^i(D) = \frac{1}{n_i}\left(h^i_1 + \cdots + h^i_{n_i}\right)$ is the average of labeled node embedding in training set $D$, where $n_i$ is the number of nodes in class $i$. $C^i$ follows the distribution $C^i \sim N\left(\mu^i, \frac{1}{n_i}\Lambda^i\right)$. Similarly, we denote $e^i = C^i - \mu^i$ and it follows that $e^i \sim N\left(0, \frac{1}{n_i}\Lambda^i\right)$.

**Variance for Semi-supervised Node Classification.** Under the above assumption, we can write the variance generated by different sampling training set explicitly. For a node $x$ that belongs to class $j$, the variance can be written:

$$\text{Var}(x) = \sum_{i=1}^{c} \mathbb{E}_D\left[\left(h^T(x)C^i - \mathbb{E}_D\left[h^T(x)C^i\right]\right)^2\right]$$

$$= \sum_{i=1}^{c} \mathbb{E}_{e^i}\left[\left(h^T(x)(\mu^i + e^i) - h^T(x)\mu^i\right)^2\right] \quad (2)$$

$$= \sum_{i=1}^{c} \mathbb{E}_{e^i}\left[\left(h^T(x)e^i\right)^2\right] = \sum_{i=1}^{c} \frac{1}{n_i}h^T(x)\Lambda^i h(x)$$

The last equation is derived from the variance of multivariate normal distribution. The variance over the whole graph $\sum_{i=1}^{c} \mathbb{E}_x\left[\mathrm{Var}(x)\right] = \sum_{i=1}^{c} \mathbb{E}_x\left[\frac{1}{n_i} h^T(x)\Lambda^i h(x)\right]$ is the expectation of Equation 2 on node $x$.

**Variance and Imbalance.** In Equation 2, we notice that the variance of a specific class $i$ is proportional to $\frac{1}{n_i}$. This reveals a relation between *imbalance* and variance. If we assume $\mathbb{E}_x[h^T(x)\Lambda^i h(x)]$ is the same for different $i$, then the following theorem holds:

**Theorem 1** *Under the condition that $\sum_i n_i$ is a constant, the variance $\sum_{i=1}^{c} \mathbb{E}_x\left[\frac{1}{n_i} h^T(x)\Lambda^i h(x)\right]$ reach its minimum when all $n_i$ equal.*

The proof is include in Appendix B.1. As the ratio of imbalance increases, the minority class exhibits a smaller sample size $n_i$, which consequently makes a greater contribution to the overall variance. This result gives an new perspective of explanation about why the model shows poor performance under imbalance training, that variance will increase as the training set becomes imbalanced. To substantiate our analysis, we conducted an experimental study to show the relation between imbalance and the variance. The result is shown in Figure 1.

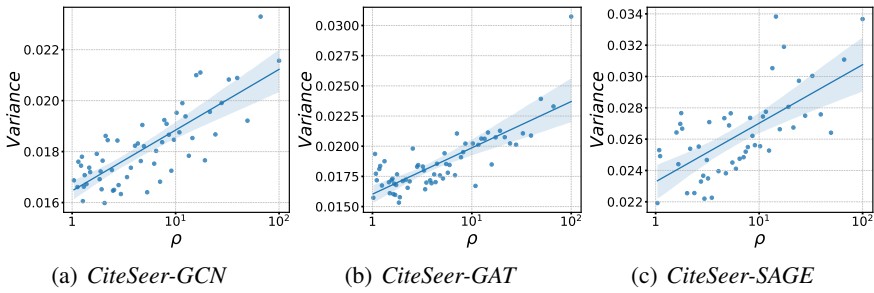

(a) *CiteSeer-GCN*          (b) *CiteSeer-GAT*          (c) *CiteSeer-SAGE*

Figure 1: We examine the alteration in variance concerning node classification as the imbalance ratio increases on and plot the regression curves for variance and imbalance ratio. We conduct this experiment using a fixed number of training set nodes but different ones, to mitigate the influence of the number of training set nodes on variance. Detailed experimental setup is in Appendix D.2.

## 3.2 Variance Regularization

**Optimize Variance during Training.** Both our analysis and the experimental results show that the increase of variance is a reason for the deterioration of performance for imbalance training set. Therefore, We propose to estimate the variance and use it as a regularization during training. We decompose the model into two parts. A GNN $E$ is utilized as a feature extractor, and a classifier $P$ predict the probability based on the feature extracted by GNN. We define the *variance of $P$* conditioning on $E$ of as Equation 2 evaluated for a fixed GNN $E$. Then we can take this as regularization. By optimizing this term, we allow the model to automatically search for the feature extractor that yields the smallest variance during the optimization process.

**Estimate the Expectation with Labeled Nodes.** The most challenging aspect lies in the estimation of variance, as we lack access to other training sets. Therefore, we propose Lemma 1 to estimate the variance on training set with the variance on labeled data. The proof is included in Appendix B.2.1.

**Lemma 1** *Under the above assumption for $h^i \sim N(\mu^i, \Lambda^i)$, $C^i \sim N\left(\mu^i, \frac{1}{n_i}\Lambda^i\right)$, minimizing the $\sum_{i=1}^{c} \mathbb{E}_x\left[\mathrm{Var}(x)\right]$ is equivalent to minimizing Equation 3:*

$$\frac{1}{N}\sum_{x\in G}\sum_{i=1}^{c}\left(h(x)^T \frac{1}{\sqrt{2n_i}}\left(h_1^i - h_2^i\right)\right)^2 = \frac{1}{N}\sum_{k=1}^{n_j}\sum_{j=1}^{c}\sum_{i=1}^{c}\left((\mu_k^j + \epsilon_k^j)^T \frac{1}{\sqrt{2n_i}}\left(\epsilon_1^i - \epsilon_2^i\right)\right)^2 . \quad (3)$$

**Estimate the Expectation with Unlabeled Nodes.** Lemma 1 allows us to replace the sampling on training set with sampling on labeled node pairs $(h_1^i, h_2^i)$ of class $i$. However, to evaluate Equation 3 it still needs to have access to embedding pair $(h_1^i, h_2^i)$ that belong to the same class $i$. Since labels are scarce, such node pairs is difficult to obtain. To make our algorithm more practical, a way of utilizing unlabeled nodes for estimating Equation 3 is required. We accomplish this goal through two steps. In the first step, we use graph augmentation to obtain pseudo embedding pair $(h_1, h_2)$. Graph augmentation is a typical used technique in graph contrastive learning. It generate new view $G'$ of a graph $G$ by adding noise to the graph structure. For a node $x$ in two different views $G, G'$, the resulting embedding $h, h'$ can be seen as that of two different nodes. Thus, they can be used to replace true embedding pair in Equation 3.

However, these pseudo node pairs give no information about which class it belongs to. Thus it prohibits us from assign proper $\frac{1}{\sqrt{2n_i}}$ for different $i$ in Equation 3. It is such coefficients that compensate on the minority class, so it's crucial to reintroduce the message about class number for constructing variance regularization. Therefore, in the second step, we use the class center $C^i$ to replace $h$ in Equation 3, as shown in Equation 4. Since $C^i = \mu^i + e^i$ and the variance of $e^i$ is proportional to $\frac{1}{n_i}$, using Equation 4 to estimate Lemma 1 can apply the similar compensatory for the minority class. More mathematical details are presented in Appendix B.2.1.

$$
\frac{1}{N} \sum_{x \in G} \sum_{i=1}^{c} \left( (C^i)^T h(x) - (C^{i'})^T h'(x) \right)^2
$$
$$
= \frac{1}{N} \sum_{k=1}^{n_j} \sum_{j=1}^{c} \sum_{i=1}^{c} \left( (\mu^i)^T (\epsilon_k^j - \epsilon_k^{j'}) + (e^i)^T \epsilon_k^j - (e^{i'})^T \epsilon_k^{j'} \right)^2
\tag{4}
$$

**From the Viewpoint of Graph Sampling.** We notice that an alternative perspective can be adopted to interpret this regularization term. Given a graph data $G$, we should acknowledge that there exists some noise in its structure or feature, as a result of measuring. This randomness also attribute to the variance of classification. Assume each graph $G$ in our dataset is a sample drawn from an underlying true graph $\tilde{G}$. Denote the classification result trained on $G$ as $f(x; G)$. Then the variance on $G$ can be written as

$$
\text{Var}_G = \mathbb{E}_{x \in G} \left[ \mathbb{E}_G \left[ [f(x; G) - \mathbb{E}_G[f(x; G)]^2] \right] \right]
\tag{5}
$$

Taking $f(x) = h^T(x)C$, it is straitforward to show that Equation 4 and Equation 5 are same.

Despite the algorithm obtained through this new interpretation being identical to the previous one, it is worth noting that the two interpretations differ significantly. The former incorporates information from unlabeled nodes and assumes the variance originates from selecting different nodes as the training set. Graph augmentation facilitates the sampling of node pairs belonging to the same class. The latter, on the other hand, considers the same training set, but the variance originates from the entire graph's information. Graph augmentation is used to simulate the process of sampling the training graph from the underlying true graph.

## 4 The Final Algorithm

In this section, we introduce our ReVar (**Re**gularize **Var**iance) framework which is based on previous theoretical frameworks for optimizing model variance. We explain our innovative approach to approximating the model's variance and using it as a regularization term in our algorithm in Section 4.1. In Section 4.2, we discuss how we integrate the GCL framework with a new contrastive learning term designed for semi-supervised learning tasks. Lastly, we present the final formulation of our optimization objective in Section 4.3, which optimizes both the cross-entropy loss function and our variance-constrained regularization term, providing a comprehensive approach to improve the performance of semi-supervised imbalanced node classification tasks.

### 4.1 Variance-constrained Optimization with Adaptive Regularization

To initiate, we elucidate the methodology underpinning the computation of Equation 4. Initially, we calculate the class center $C_i$ corresponding to each class. Following this, we systematically construct

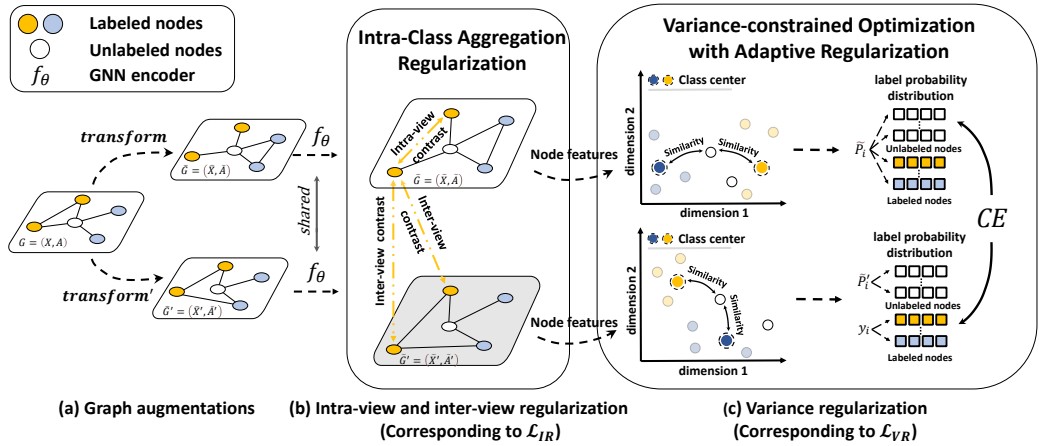

**(a) Graph augmentations**

**(b) Intra-view and inter-view regularization**
**(Corresponding to $\mathcal{L}_{IR}$)**

**(c) Variance regularization**
**(Corresponding to $\mathcal{L}_{VR}$)**

Figure 2: Overall pipeline of ReVar. (a) Two different views of the graph $\tilde{\mathbf{G}}, \tilde{\mathbf{G}}'$ are obtained by graph augmentation $transform$, and are subsequently fed into GNN encoder $f_\theta$. (b) Intra-class and inter-class representations are aggregated, which means, for labeled nodes, it's positive samples not only belong to the same class in both view but also in the other view. (c) Variance is estimated by Equation 4. Specifically, the label probability distribution is computed for each node in two views based on it's similarity with each class center. And the difference between two probability distributions is used to approximate the model's variance and also optimized as one term in the loss function.

a probability distribution reference matrix, denoted by $\mathcal{S} := [C_1; C_2; \ldots; C_k]$. It's worth noting that the assemblage of class centers is represented by $C := (C_1; C_2; \ldots; C_k)$.

Given a node $i$ with its associated embedding $h_i$, the label probability distribution $\pi_i$ pertaining to node $i$ is ascertainable via the subsequent equation:

$$\pi_i^j = \frac{\exp\left(\text{sim}\left(\text{h}_i, C_j\right)/\tau\right)}{\sum_{l=1}^k \exp\left(\text{sim}\left(\text{h}_i, C_l\right)/\tau\right)} \tag{6}$$

In the aforementioned equation, $\pi_i^j$ signifies the $j$-th dimensional component of $\pi_i$, $\text{sim}(\cdot, \cdot)$ denotes the function to compute the cosine similarity between a pair of vectors, and $\tau$ is a specified temperature hyperparameter.

Subsequent to this, for every node $i \in V$, we determine the label probability distribution. Distinctively, $\tilde{\pi}_i$ and $\tilde{\pi}_i'$ are derived from $\tilde{G}$ and $\tilde{G}'$, respectively. The former acts as the prediction distribution, whereas the latter represents the target distribution. Our approach then aims to minimize the cross-entropy between these two distributions.

It's paramount to underscore that for each labeled node within $\tilde{G}'$, its one-hot label vector is directly assigned as $\tilde{\pi}_i' = y_i$ for all $i \in V_L$. This diverges from the procedure of deducing the predicted class distribution as described in Equation 6. This methodology is purposed to optimally leverage the extant label information.

However, a simplistic minimization of the aforementioned loss might engender confirmation bias. This is attributed to the potentially inaccurate $\tilde{\pi}_i'$ estimated for the unlabeled nodes (i.e., $V_U$), an aspect that can deleteriously impact the efficacy of pseudo-labeling-oriented semi-supervised methodologies. To address this concern, we introduce a confidence-based label-guided consistency regularization [41, 30, 48], as detailed below:

$$\mathcal{L}_{\text{VR}} = \frac{1}{|V_{\text{conf}}|} \sum_{i \in V_{\text{conf}}} CE\left(\tilde{\pi}_i', \tilde{\pi}_i\right) + \frac{1}{|V_L|} \sum_{i \in V_L} CE\left(y_i, \tilde{\pi}_i\right) \tag{7}$$

where $V_{\text{conf}} = \{v_i \mid \mathbf{1}_{\{\max(\tilde{\pi}_i') > v\}} = 1, \forall i \in V_U\}$ represents the set of nodes with confident predictions, $v$ is the threshold for determining whether a node has a confident prediction, and $\mathbf{1}_{\{\ldots\}}$ is an indicator function.

In the context of confidence determination for $\tilde{\pi}'_i$, we employ a criterion wherein a value is appraised as confident if its maximum constituent surpasses a stipulated threshold. We postulate that a heightened threshold, represented by $v$, acts as a safeguard against confirmation bias. This stratagem ensures that only high-caliber target distributions, specifically $\tilde{\pi}'_i$, have a pivotal influence on Equation 7.

## 4.2 Intra-Class Aggregation Regularization

In this section, we propose an extension to the concept of graph contrastive learning that emphasizes the invariance of node representations in semi-supervised scenarios. To achieve this, we partition the nodes into two groups: labeled and unlabeled. Specifically, for the unlabeled nodes, we learn node representation invariance through the use of positive examples exclusively. For each labeled node in both views, its positive sample no longer includes itself in the other view, but instead encompasses all labeled nodes in both views belonging to the same class. Through this approach, labeled nodes belonging to the same class can be aggregated, enabling the model to learn that representation invariance now applies not to a single node, but to the complete class representation.

$$\mathcal{L}_{\text{IR}} = -\frac{1}{|V_U|} \sum_{\text{h}_i, \text{h}'_i \in V_U} \text{sim} \left( \text{h}_i \cdot \text{h}'_i \right) - \frac{1}{N_{all}} \left( \sum_{l=1}^{k} \sum_{\text{h}_i, \text{h}'_j \in \mathbf{C}_l} \text{sim} \left( \text{h}_i \cdot \text{h}'_j \right) + \sum_{l=1}^{k} \sum_{\substack{\text{h}_i, \text{h}_j \in \mathbf{C}_l \\ i \neq j}} \text{sim} \left( \text{h}_i \cdot \text{h}_j \right) \right) \tag{8}$$

where $N_{all} = \sum_{i=1}^{k} |\mathbf{C}_i| \left( |\mathbf{C}_i| - 1 \right)$, $\text{h}_i$ and $\text{h}'_i$ represent feature embedding in $\tilde{G}$ and $\tilde{G}'$ respectively for node $i$. To the best of our knowledge, we are the first to introduce label information into the contrastive learning paradigm to obtain invariant representations within the context of intra-class variation and address the challenge of semi-supervised imbalanced node classification. Notably, our approach only utilizes positive examples, which significantly reduces the model's training complexity and enhances its scalability and generalization capabilities.

## 4.3 Objective Function

To derive our ultimate objective function, we incorporate both $\mathcal{L}_{\text{VR}}$ and $\mathcal{L}_{\text{IR}}$. These are weighted by coefficients $\lambda_1$ and $\lambda_2$, respectively. Formally, the amalgamation can be represented as:

$$\mathcal{L}_{\text{composite}} = \lambda_1 \mathcal{L}_{\text{VR}} + \lambda_2 \mathcal{L}_{\text{IR}} + \mathcal{L}_{\text{sup}} \tag{9}$$

Additionally, we introduce the cross-entropy loss denoted by $\mathcal{L}_{\text{sup}}$, which is established over a collection of labeled nodes, denominated as $V_L$.

# 5 Experiment

## 5.1 Experimental Setups

**Datasets and Baselines.** We have demonstrated the efficacy of our method on five commonly used benchmark datasets across various imbalance scenarios. For the conventional setting ($\rho$=10) of imbalanced node classification in [50, 24, 31], we conducted experiments on Cora, CiteSeer, Pubmed, and Amazon-Computers. More precisely, we select half of the classes as the minority classes and randomly convert labeled nodes into unlabeled ones until the training set reaches an imbalance ratio of $\rho$. To be more precise, for the three citation networks, we adopt the standard splits proposed by [44] as our initial splits, with an initial imbalance ratio of $\rho$. In order to better reflect real-world scenarios, we conducted representative experiments on the naturally imbalanced datasets Amazon-Computers and Coauthor-CS. For this setting, we utilize random sampling to construct a training set that adheres to the true label distribution of the entire graph. The comprehensive experimental settings, including the evaluation protocol and implementation details of our algorithm, are explicated in Appendix E. For baselines, we evaluate our method against classic techniques, including cross-entropy loss with re-weighting [13], PC Softmax [12], and Balanced Softmax [26], as well as state-of-the-art methods for imbalanced node classification, such as GraphSMOTE [50], GraphENS [24], ReNode [6], and TAM [31]. See Appendix E for implementation details of the baselines.

**GNN Encoder Selection.** As the foundational structures for our GNN encoder, we harness architectures such as GCN [16], GAT [35], and GraphSAGE [9].

## 5.2 Main Results

**Experimental Results Under Traditional Semi-supervised Settings.** Table 1 presents the averaged balanced accuracy (bAcc.) and F1 score, along with their standard errors, obtained by the baselines and ReVar on three class-imbalanced node classification benchmark datasets, where $\rho = 10$. The outcomes showcase the superiority of ReVar in comparison to the existing methods. Our proposed technique exhibits a consistent and significant outperformance of the state-of-the-art approaches across all three datasets and three base models. In all cases, ReVar achieves a decisive advantage that underscores its efficacy in addressing the challenge of class imbalance in node classification. Due to space limitations, the experimental results and analysis on Cora are presented in the Appendix C.

Table 1: Comparison of our method ReVar and other baselines on *three benchmark datasets*. Experimental results are measured by averaged balanced accuracy (bAcc.,%) and F1-score (%) with the standard errors over 5 repetitions on three GNN architectures. Highlighted are the top **first** and **second**. $\Delta$ is the margin by which our method leads state-of-the-art method.

| | Dataset | *CiteSeer-Semi* | | *PubMed-Semi* | | *Computers-Semi* | |
|---|---|---|---|---|---|---|---|
| | **Imbalance Ratio ($\rho = 10$)** | bAcc. | F1 | bAcc. | F1 | bAcc. | F1 |
| **GCN** | Vanilla | $38.72 \pm 1.88$ | $28.74 \pm 3.21$ | $65.64 \pm 1.72$ | $56.97 \pm 3.17$ | $80.01 \pm 0.71$ | $71.56 \pm 0.81$ |
| | Re-Weight | $44.69 \pm 1.78$ | $38.61 \pm 2.37$ | $69.06 \pm 1.84$ | $64.08 \pm 2.97$ | $80.93 \pm 1.30$ | $73.99 \pm 2.20$ |
| | PC Softmax | $50.18 \pm 0.55$ | $46.14 \pm 0.14$ | $72.46 \pm 0.80$ | $70.27 \pm 0.94$ | $81.54 \pm 0.76$ | $73.30 \pm 0.51$ |
| | GraphSMOTE | $44.87 \pm 1.12$ | $39.20 \pm 1.62$ | $67.91 \pm 0.64$ | $62.68 \pm 1.92$ | $79.48 \pm 0.47$ | $72.63 \pm 0.76$ |
| | BalancedSoftmax | $55.52 \pm 0.97$ | $53.74 \pm 1.42$ | $73.73 \pm 0.89$ | $71.53 \pm 1.06$ | $81.46 \pm 0.74$ | $74.31 \pm 0.51$ |
| | + TAM | $56.73 \pm 0.71$ | $56.15 \pm 0.78$ | $74.62 \pm 0.97$ | $72.25 \pm 1.30$ | $82.36 \pm 0.67$ | $72.94 \pm 1.43$ |
| | Renode | $43.47 \pm 2.22$ | $37.52 \pm 3.10$ | $71.40 \pm 1.42$ | $67.27 \pm 2.96$ | $81.89 \pm 0.77$ | $73.13 \pm 1.60$ |
| | + TAM | $46.20 \pm 1.17$ | $39.96 \pm 2.76$ | $72.63 \pm 2.03$ | $68.28 \pm 3.30$ | $80.36 \pm 1.19$ | $72.51 \pm 0.68$ |
| | GraphENS | $56.57 \pm 0.98$ | $55.29 \pm 1.33$ | $72.13 \pm 1.04$ | $70.72 \pm 1.07$ | $82.40 \pm 0.39$ | $74.26 \pm 1.05$ |
| | + TAM | $58.01 \pm 0.68$ | $56.32 \pm 1.03$ | $74.14 \pm 1.42$ | $72.42 \pm 1.39$ | $81.02 \pm 0.99$ | $70.78 \pm 1.72$ |
| | **ReVar** | $65.28 \pm 0.51$ | $64.91 \pm 0.51$ | $79.20 \pm 0.72$ | $78.45 \pm 0.46$ | $84.67 \pm 0.17$ | $80.25 \pm 0.87$ |
| | $\Delta$ | $+7.27(12.53\%)$ | $+8.59(15.25\%)$ | $+5.06(6.82\%)$ | $+6.03(8.33\%)$ | $+2.27(2.75\%)$ | $+5.94(7.99\%)$ |
| **GAT** | Vanilla | $38.84 \pm 1.13$ | $31.25 \pm 1.64$ | $64.60 \pm 1.64$ | $55.24 \pm 2.80$ | $79.04 \pm 1.60$ | $70.00 \pm 2.50$ |
| | Re-Weight | $45.47 \pm 2.35$ | $40.60 \pm 2.98$ | $68.10 \pm 2.85$ | $63.76 \pm 3.54$ | $80.38 \pm 0.66$ | $69.99 \pm 0.76$ |
| | PC Softmax | $50.78 \pm 1.66$ | $48.56 \pm 2.08$ | $72.88 \pm 0.83$ | $71.09 \pm 0.89$ | $79.43 \pm 0.94$ | $71.33 \pm 0.86$ |
| | GraphSMOTE | $45.68 \pm 0.93$ | $38.96 \pm 0.97$ | $67.43 \pm 1.23$ | $61.97 \pm 2.54$ | $79.38 \pm 1.97$ | $69.76 \pm 2.31$ |
| | BalancedSoftmax | $54.78 \pm 1.25$ | $51.83 \pm 2.11$ | $72.30 \pm 1.20$ | $69.30 \pm 1.79$ | $82.02 \pm 1.19$ | $72.94 \pm 1.54$ |
| | + TAM | $56.30 \pm 1.25$ | $53.87 \pm 1.14$ | $73.50 \pm 1.24$ | $71.36 \pm 1.99$ | $75.54 \pm 2.09$ | $66.69 \pm 1.44$ |
| | Renode | $44.48 \pm 2.06$ | $37.93 \pm 2.87$ | $69.93 \pm 2.10$ | $65.27 \pm 2.90$ | $76.01 \pm 1.08$ | $66.72 \pm 1.42$ |
| | + TAM | $45.12 \pm 1.41$ | $39.29 \pm 1.79$ | $70.66 \pm 2.13$ | $66.94 \pm 3.54$ | $74.30 \pm 1.13$ | $66.13 \pm 1.75$ |
| | GraphENS | $51.45 \pm 1.28$ | $47.98 \pm 2.08$ | $73.15 \pm 1.24$ | $71.90 \pm 1.03$ | $81.23 \pm 0.74$ | $71.23 \pm 0.42$ |
| | + TAM | $56.15 \pm 1.13$ | $54.31 \pm 1.68$ | $73.45 \pm 1.07$ | $72.10 \pm 0.36$ | $81.07 \pm 1.03$ | $71.27 \pm 1.98$ |
| | **ReVar** | $66.04 \pm 0.66$ | $65.70 \pm 0.69$ | $77.85 \pm 0.76$ | $77.08 \pm 0.69$ | $86.37 \pm 0.02$ | $82.35 \pm 0.02$ |
| | $\Delta$ | $+9.89(17.61\%)$ | $+11.39(20.97\%)$ | $+4.40(5.99\%)$ | $+4.98(6.91\%)$ | $+4.35(5.30\%)$ | $+9.41(12.90\%)$ |
| **SAGE** | Vanilla | $43.18 \pm 0.52$ | $36.66 \pm 1.25$ | $68.68 \pm 1.51$ | $64.16 \pm 2.38$ | $72.36 \pm 2.39$ | $64.32 \pm 2.21$ |
| | Re-Weight | $46.17 \pm 1.32$ | $40.13 \pm 1.68$ | $69.89 \pm 1.60$ | $65.71 \pm 2.31$ | $76.08 \pm 1.14$ | $65.76 \pm 1.40$ |
| | PC Softmax | $50.66 \pm 0.99$ | $47.48 \pm 1.66$ | $71.49 \pm 0.94$ | $70.23 \pm 0.67$ | $74.63 \pm 3.01$ | $66.44 \pm 4.04$ |
| | GraphSMOTE | $42.73 \pm 2.87$ | $35.18 \pm 1.75$ | $66.63 \pm 0.65$ | $61.97 \pm 2.54$ | $71.85 \pm 0.98$ | $68.92 \pm 0.73$ |
| | BalancedSoftmax | $51.74 \pm 2.32$ | $49.01 \pm 3.16$ | $71.36 \pm 1.81$ | $69.66 \pm 1.81$ | $73.67 \pm 1.11$ | $65.23 \pm 2.44$ |
| | + TAM | $51.93 \pm 2.19$ | $48.67 \pm 3.25$ | $72.28 \pm 1.47$ | $71.02 \pm 1.31$ | $77.00 \pm 2.93$ | $70.85 \pm 2.28$ |
| | Renode | $48.65 \pm 1.37$ | $44.25 \pm 2.20$ | $71.37 \pm 1.33$ | $67.78 \pm 1.38$ | $77.37 \pm 0.74$ | $68.42 \pm 1.81$ |
| | + TAM | $48.39 \pm 1.76$ | $43.56 \pm 2.31$ | $71.25 \pm 1.07$ | $68.69 \pm 0.98$ | $74.87 \pm 2.25$ | $66.87 \pm 2.52$ |
| | GraphENS | $53.51 \pm 0.78$ | $51.42 \pm 1.19$ | $70.97 \pm 0.78$ | $70.00 \pm 1.22$ | $82.57 \pm 0.50$ | $71.95 \pm 0.51$ |
| | + TAM | $54.69 \pm 1.12$ | $53.56 \pm 1.86$ | $73.61 \pm 1.35$ | $72.50 \pm 1.58$ | $82.17 \pm 0.93$ | $72.46 \pm 1.00$ |
| | **ReVar** | $60.48 \pm 0.88$ | $57.99 \pm 1.54$ | $77.72 \pm 1.06$ | $76.01 \pm 1.20$ | $83.50 \pm 0.02$ | $76.48 \pm 0.05$ |
| | $\Delta$ | $+5.79(10.59\%)$ | $+4.53(8.46\%)$ | $+4.11(5.58\%)$ | $+3.51(4.84\%)$ | $+0.93(1.13\%)$ | $+4.02(5.55\%)$ |

**Experimental Results On Naturally Imbalanced Datasets.** Our model was evaluated on two naturally imbalanced datasets, Coauthor-CS ($\rho \approx 41.0$) and Amazon-Computers ($\rho \approx 17.7$), both of which exhibit imbalanced unlabeled data (as indicated in Table 7). The construction of the training, validation, and testing sets is detailed in Appendix E. Experimental results are presented in Table

Table 2: Comparison of our method ReVar and other baselines on *CS-Random*. Highlighted are the top **first** and **second**. $\Delta$ is the margin by which our method leads state-of-the-art method.

| Dataset(*CS-Random*) | GCN | | GAT | | SAGE | |
|---|---|---|---|---|---|---|
| **Imbalance Ratio**($\rho = 41.00$) | bAcc. | F1 | bAcc. | F1 | bAcc. | F1 |
| Vanilla | $84.85 \pm 0.16$ | $87.12 \pm 0.14$ | $82.47 \pm 0.36$ | $84.21 \pm 0.31$ | $83.76 \pm 0.27$ | $86.22 \pm 0.19$ |
| Re-Weight | $87.42 \pm 0.17$ | $88.70 \pm 0.10$ | $83.55 \pm 0.39$ | $84.73 \pm 0.32$ | $85.76 \pm 0.24$ | $87.32 \pm 0.16$ |
| PC Softmax | $88.36 \pm 0.12$ | $88.94 \pm 0.04$ | $85.22 \pm 0.31$ | $85.54 \pm 0.33$ | $87.18 \pm 0.14$ | $88.00 \pm 0.19$ |
| GraphSMOTE | $85.76 \pm 1.73$ | $87.31 \pm 1.32$ | $84.65 \pm 1.32$ | $85.63 \pm 1.01$ | $85.76 \pm 1.98$ | $87.34 \pm 0.98$ |
| BalancedSoftmax | $87.72 \pm 0.07$ | $88.67 \pm 0.07$ | $84.38 \pm 0.20$ | $84.53 \pm 0.41$ | $86.78 \pm 0.10$ | $88.05 \pm 0.09$ |
| + TAM | $88.22 \pm 0.11$ | $89.22 \pm 0.08$ | $85.48 \pm 0.24$ | $85.77 \pm 0.50$ | $87.83 \pm 0.13$ | $88.77 \pm 0.07$ |
| Renode | $87.53 \pm 0.11$ | $88.91 \pm 0.06$ | $85.98 \pm 0.19$ | $86.97 \pm 0.09$ | $86.13 \pm 0.10$ | $87.89 \pm 0.09$ |
| + TAM | $87.55 \pm 0.06$ | $89.03 \pm 0.05$ | $86.61 \pm 0.30$ | $87.42 \pm 0.24$ | $85.21 \pm 0.33$ | $87.01 \pm 0.31$ |
| GraphENS | $85.97 \pm 0.29$ | $86.68 \pm 0.20$ | $85.86 \pm 0.19$ | $86.51 \pm 0.32$ | $85.39 \pm 0.26$ | $86.41 \pm 0.24$ |
| + TAM | $86.34 \pm 0.12$ | $87.36 \pm 0.08$ | $86.29 \pm 0.20$ | $87.28 \pm 0.13$ | $85.99 \pm 0.13$ | $87.25 \pm 0.07$ |
| **ReVar** | $88.44 \pm 0.16$ | $89.54 \pm 0.11$ | $87.33 \pm 0.04$ | $88.33 \pm 0.06$ | $90.11 \pm 0.11$ | $91.18 \pm 0.11$ |
| $\Delta$ | $+ 0.08$(0.09%) | $+ 0.32$(0.36%) | $+ 0.72$(0.83%) | $+ 0.91$(1.04%) | $+ 2.28$(2.60%) | $+ 2.41$(2.71%) |

4 and Table 2. Notably, ReVar outperformed other methods consistently on these two datasets, a significant finding given the challenging nature of the imbalanced data in these datasets.

## 5.3 Further Analysis of ReVar

**Analysis of Each Component in the Loss Function.** As shown in Figure 3(a), each component of our loss function can bring performance improvements. In particular, in all three settings in the figure, our full loss achieves the best F1 scores. In all cases, VR loss improves the performance of the model, proving our hypothesis that can approximate the variance of the model to improve the performance of the model. More analysis is presented in Appendix D.

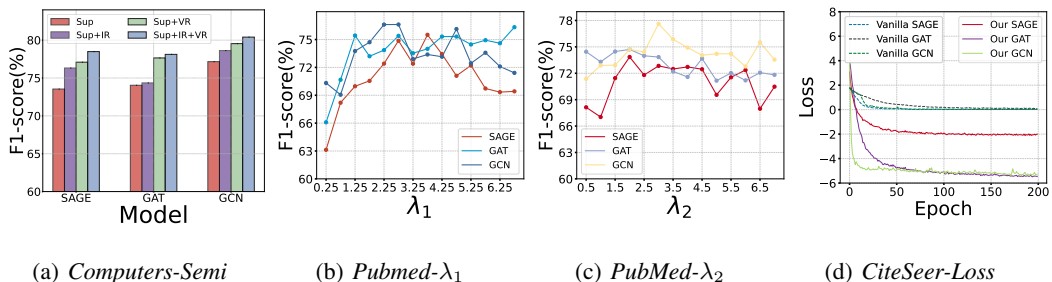

| (a) *Computers-Semi* | (b) *Pubmed-$\lambda_1$* | (c) *PubMed-$\lambda_2$* | (d) *CiteSeer-Loss* |
|---|---|---|---|

Figure 3: Analysis of ReVar.

**Sensitive to Hyperparameters $\lambda_1$ and $\lambda_2$.** The two intensity terms, $\lambda_1$ and $\lambda_2$, corresponding to the $\mathcal{L}_{\text{VR}}$ and $\mathcal{L}_{\text{IR}}$, respectively, is depicted in Figure 3(b), 3(c). We explore the impact of perturbations on model performance when one of the two intensity terms is held constant while the other is varied. We believe that a smaller value of $\lambda$ is inadequate to reflect the regularization of ReVar. Instead, suitable values of $\lambda$ can significantly enhance the performance of ReVar. More analysis is presented in Appendix D.

**Analysis of Test Loss and F1 Score Curves.** The present study includes a comparison of ReVar and Vanilla models(only use cross-entropy) in terms of their loss and F1 score. As shown in Figure 3(d), various models were evaluated. The ReVar model, which incorporates additional $\mathcal{L}_{\text{VR}}$ and $\mathcal{L}_{\text{IR}}$ components, demonstrated a faster convergence of loss, as evidenced by the larger slope in Figure 3(d). More analysis is presented in Appendix D.

# 6    Conclusion and Future Work

This paper presents a novel approach to address imbalanced node classification. Our method integrates imbalanced node classification and the Bias-Variance Decomposition framework, establishing a theoretical foundation that closely links data imbalance to model variance. Additionally, we employ graph augmentation techniques to estimate model variance and design a regularization term to mitigate the impact of class imbalance. We conduct exhaustive testing to demonstrate superior performance compared to state-of-the-art methods in various imbalanced scenarios. Future work includes extending ReVar and its theories to the fields of computer vision and natural language processing. Moreover, we anticipate the emergence of more effective theories and algorithms to be developed and flourish.

### Acknowledgements

This work is supported by National Natural Science Foundation of China No.U2241212, No.62276066.

### Reproducibility Statements

The model implementation and data is released at https://github.com/yanliang3612/ReVar.

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

# A Other related work

**Imbalanced Learning in Traditional ML.** Most real-world data is naturally imbalanced, presenting a significant challenge in training fair models that are not biased towards majority classes. To address this problem, various approaches have been commonly utilized. Ensemble learning [7, 18, 52, 39, 20, 3] combines the outputs of multiple weak classifiers. Data re-sampling methods [5, 10, 29, 27, 14, 37] aim to adjust the label distribution in the training set by synthesizing or duplicating samples from the minority class. Another approach tackles the imbalance issue by modifying the loss function, assigning larger weights to minority classes or adjusting the margins between different classes [53, 33, 4, 32, 42, 26, 38]. Post-hoc correction methods compensate for the imbalanced classes during the inference step, after completing the model training [14, 34, 21, 12]. Although these techniques have been extensively applied to i.i.d. data, extending them to graph-structured data poses non-trivial challenges.

**Graph Contrastive Learning.** Contrastive methods, which have proven effective for unsupervised learning in vision, have also been adapted for graph data. One notable approach is DGI [36], which presents a framework for unsupervised node-level representation learning that maximizes global mutual information. Other approaches, such as GRACE [54], GCA [55], and GraphCL [47], utilize augmented graphs to optimize the similarity between positive node pairs and minimize negative pairs. CCA-SSG [49] introduces an efficient loss function based on canonical correlation analysis, eliminating the need for negative samples. Incorporating community information, gCooL [17] enhances node representations and downstream task performance. GGD [51] simplifies the mutual information loss function by directly discriminating between two sets of node samples, resulting in faster computation and lower memory usage. These contrastive methods exhibit potential for improving unsupervised learning on graph data. However, it is important to note that our model operates within the context of semi-supervised learning, which significantly differs from the mechanisms employed by these models.

## B  Proofs

### B.1  Proofs of Theorem 1

**Theorem 1** *Under the condition that $\sum_{i=1}^{c} n_i$ is a constant, the variance $\sum_{i=1}^{c} \mathbb{E}_x \left[ \frac{1}{n_i} h^T(x) \Lambda^i h(x) \right]$ reach its minimum when all $n_i$ equal.*

*Proof*  The expression $\sum_{i=1}^{c} \mathbb{E}_x \left[ \frac{1}{n_i} h^T(x) \Lambda^i h(x) \right]$ can be equivalently expressed as $\sum_{i=1}^{c} \frac{1}{n_i} \mathbb{E}_x \left[ h^T(x) \Lambda^i h(x) \right]$. As previously assumed, the $\mathbb{E}_x [h^T(x) \Lambda^i h(x)]$ is the same for different $i$, which implies that our goal is to demonstrate that $\sum_{i=1}^{c} \frac{1}{n_i}$ is minimized when all $n_i$ are equal.

Let $m$ be $\sum_{i=1}^{c} n_i$. We wish to find the extremum of the sum of their reciprocals, which is given by

$$S = \frac{1}{n_1} + \frac{1}{n_2} + \cdots + \frac{1}{n_c}. \tag{10}$$

Using the inequality of arithmetic and harmonic means, we have

$$\frac{c}{\frac{1}{n_1} + \frac{1}{n_2} + \cdots + \frac{1}{n_c}} \leq \frac{n_1 + n_2 + \cdots + n_c}{c}, \tag{11}$$

with equality if and only if $n_1 = n_2 = \cdots = n_c$. Rearranging, we get

$$\frac{c^2}{n_1 + n_2 + \cdots + n_c} \leq S, \tag{12}$$

with equality if and only if $n_1 = n_2 = \cdots = n_c$. Since $m = n_1 + n_2 + \cdots + n_c$, we have

$$\frac{c^2}{m} \leq S, \tag{13}$$

with equality if and only if $n_1 = n_2 = \cdots = n_c = \frac{m}{c}$. Therefore, when all the $c$ numbers are equal, the sum of their reciprocals is minimized and given by $S = \frac{c^2}{m}$.

We can also use the method of Lagrange multipliers. Let $f(n_1, n_2, \ldots, a_c) = \frac{1}{n_1} + \frac{1}{n_2} + \ldots + \frac{1}{n_c}$ be the function that we want to extremize. Then, the Lagrangian is:

$$\mathcal{L}(n_1, n_2, \ldots, n_c, \lambda) = f(n_1, n_2, \ldots, n_c) + \lambda(n_1 + n_2 + \ldots + n_c - m). \tag{14}$$

Taking the partial derivatives of $\mathcal{L}$ with respect to $a_i$ and $\lambda$, we get:

$$\begin{aligned} \frac{\partial \mathcal{L}}{\partial n_i} &= -\frac{1}{n_i^2} + \lambda \\ \frac{\partial \mathcal{L}}{\partial \lambda} &= n_1 + n_2 + \ldots + n_c - m. \end{aligned} \tag{15}$$

Setting these partial derivatives to zero, we get:

$$n_1 = n_2 = \ldots = n_c = \frac{m}{c}, \lambda = \frac{c^2}{m^2}. \tag{16}$$

Thus, when the $c$ numbers are equal, their reciprocal sum is minimized and is equal to $\frac{c^2}{m}$. Moreover, since $S$ is a continuously differentiable function of $n_1, n_2, \ldots, n_c$, this extremum is a minimum.

Therefore, we have shown that the reciprocal sum of $c$ numbers with a sum of $m$ is minimized and equal to $\frac{c^2}{m}$ when the $c$ numbers are equal.

$\square$

### B.2 Proofs of Lemma 1 and More Details for Estimating the Expectation with Unlabeled Nodes (Section 3.2)

#### B.2.1 Proofs of Lemma 1

**Lemma 1** *Under the above assumption for $h^i \sim N(\mu^i, \Lambda^i)$, $C^i \sim N\left(\mu^i, \frac{1}{n_i}\Lambda^i\right)$, minimizing the $\sum_{i=1}^c \mathbb{E}_x\left[\text{Var}(x)\right]$ is equivalent to minimizing Equation 17:*

$$\frac{1}{N}\sum_{x\in G}\sum_{i=1}^c\left(h(x)^T\frac{1}{\sqrt{2n_i}}\left(h_1^i - h_2^i\right)\right)^2 = \frac{1}{N}\sum_{k=1}^{n_j}\sum_{j=1}^c\sum_{i=1}^c\left((\mu_k^j + \epsilon_k^j)^T\frac{1}{\sqrt{2n_i}}\left(\epsilon_1^i - \epsilon_2^i\right)\right)^2. \quad (17)$$

*Proof* Let $\epsilon^i$ denote $h^i - \mu^i$, which follows the distribution $\epsilon^i \sim N(0, \Lambda^i)$. Similarly, we denote $e^i = C^i - \mu^i$ and it follows that $e^i \sim N\left(0, \frac{1}{n_i}\Lambda^i\right)$.

We know

$$\text{Var}(x) = \sum_{i=1}^c \mathbb{E}_D\left[\left(h^T(x)C^i - \mathbb{E}_D\left[h^T(x)C^i\right]\right)^2\right]$$

$$= \sum_{i=1}^c \mathbb{E}_{e^i}\left[\left(h^T(x)(\mu^i + e^i) - h^T(x)\mu^i\right)^2\right] \quad (18)$$

$$= \sum_{i=1}^c \mathbb{E}_{e^i}\left[\left(h^T(x)e^i\right)^2\right] = \sum_{i=1}^c \frac{1}{n_i}h^T(x)\Lambda^i h(x),$$

so we get

$$\sum_{i=1}^c \mathbb{E}_x\left[\text{Var}(x)\right] = \sum_{i=1}^c \mathbb{E}_x\left[\frac{1}{n_i}h^T(x)\Lambda^i h(x)\right]. \quad (19)$$

Motivated by stochastic gradient descent, we opt to sample an $e^i$ on each occasion and compute, as opposed to directly calculating the expectation $\mathbb{E}_{D\subset G}$.

At present, we remain uncertain about how to sample the noise term $e^i$ associated with the class center $C_i$. We put forth a proposition to estimate the class center noise $e$ utilizing the feature noise $\epsilon$. Under the supposition that when $v \in C_k$, the $j_{th}$ element of the feature $f(v)$ adheres to a Gaussian distribution:

$$\epsilon^i \sim N\left(0, \Lambda^i\right), \ e^i \sim N\left(0, \frac{1}{n_i}\Lambda^i\right). \quad (20)$$

Consequently, multiplying the noise term $\epsilon^i$ in Equation 20 by $\frac{1}{\sqrt{n_i}}$ yields a random variable that exhibits an identical distribution to $e^i$:

$$\frac{1}{\sqrt{n_i}}\epsilon^i \sim N\left(0, \frac{1}{n_i}\Lambda^i\right) \quad (21)$$

Moreover, how might we compute $\varepsilon_i$? In practical scenarios, we merely possess the feature $h$. However, we are able to calculate the disparity between two features, $h_1^i$ and $h_2^i$, originating from the same class $i$:

$$\frac{1}{\sqrt{2n_i}}\left(h_1^i - h_2^i\right) = \frac{1}{\sqrt{2n_i}}(\epsilon_1 - \epsilon_2) \sim N\left(0, \frac{1}{n_i}\Lambda^i\right). \quad (22)$$

Incorporating this into Equation 19, the loss is expressed as:

$$\frac{1}{N}\sum_{x\in G}\sum_{i=1}^c\left(h(x)^T\frac{1}{\sqrt{2n_i}}\left(h_1^i - h_2^i\right)\right)^2. \quad (23)$$

$\square$

### B.2.2 More Details for Estimating the Expectation with Unlabeled Nodes (Section 3.2)

Lemma 1 suggests that $\sum_{i=1}^{c} \mathbb{E}_x [\mathrm{Var}(x)]$ can be estimated by sampling labeled node pairs $(h_1^i, h_2^i)$ from the same class $i$. However, due to the scarcity of data in the minority class, this can often be challenging. In Section 3.2, we address this issue by utilizing random data augmentation and defining nodes from different views as members of the same class, enabling us to sample node pairs for Equation 3 and perform embedding subtraction.

However, as we mentioned, the pseudo node pairs lack information regarding their class membership, making it impossible to assign appropriate $\frac{1}{\sqrt{2n_i}}$ values for different $i$ in Equation 3. These coefficients play a crucial role in compensating for the minority class, making it imperative to reintroduce information about the class number when constructing variance regularization. Therefore, in the second step, we replace $h$ in Equation 3 with the class center $C^i$, as demonstrated in Equation 4. As $C^i = \mu^i + e^i$ and the variance of $e^i$ is proportional to $\frac{1}{n_i}$, using Equation 4 to estimate Lemma 1 can provide similar compensatory effects for the minority class. We present the proof for the aforementioned propositions below.

*Proof*   Firstly, the $\sum_{i=1}^{c} \mathbb{E}_x [\mathrm{Var}(x)]$ can be decomposed as follows:

$$
\sum_{i=1}^{c} \mathbb{E}_x [\mathrm{Var}(x)] = \frac{1}{N} \sum_{x \in G} \sum_{i=1}^{c} \left( h(x)^T \frac{1}{\sqrt{2n_i}} \left( h_1^i - h_2^i \right) \right)^2
$$

$$
= \frac{1}{N} \sum_{k=1}^{n_j} \sum_{j=1}^{c} \sum_{i=1}^{c} \left( \left( \mu_k^j + \epsilon_k^j \right)^T \frac{1}{\sqrt{2n_i}} \left( h_1^i - h_2^i \right) \right)^2
$$

$$
= \frac{1}{N} \sum_{k=1}^{n_j} \sum_{j=1}^{c} \sum_{i=1}^{c} \left( (\mu_k^j)^T \frac{1}{\sqrt{2n_i}} \left( \epsilon_1^i - \epsilon_2^i \right) + (\epsilon_k^j)^T \frac{1}{\sqrt{2n_i}} \left( \epsilon_1^i - \epsilon_2^i \right) \right)^2
$$

$$
= \frac{1}{2N} \sum_{k=1}^{n_j} \sum_{j=1}^{c} \sum_{i=1}^{c} \left( (\mu_k^j)^T \frac{1}{\sqrt{n_i}} \left( \epsilon_1^i - \epsilon_2^i \right) + (\epsilon_k^j)^T \frac{1}{\sqrt{n_i}} \left( \epsilon_1^i - \epsilon_2^i \right) \right)^2
$$

$$
= \underbrace{\frac{1}{2N} \sum_{k=1}^{n_j} \sum_{j=1}^{c} \sum_{i=1}^{c} \left( \frac{1}{n_i} \left[ (\mu_k^j)^T \left( \epsilon_1^i - \epsilon_2^i \right) \right]^2 \right)}_{\mathcal{T}_1} + \underbrace{\frac{1}{2N} \sum_{k=1}^{n_j} \sum_{j=1}^{c} \sum_{i=1}^{c} \left( \frac{1}{n_i} \left[ (\epsilon_k^j)^T \left( \epsilon_1^i - \epsilon_2^i \right) \right]^2 \right)}_{\mathcal{T}_2}
$$

$$
+ \underbrace{\frac{1}{N} \sum_{k=1}^{n_j} \sum_{j=1}^{c} \sum_{i=1}^{c} \left( \frac{1}{n_i} \left[ (\mu_k^j)^T \left( \epsilon_1^i - \epsilon_2^i \right) (\epsilon_k^j)^T \left( \epsilon_1^i - \epsilon_2^i \right) \right] \right)}_{\mathcal{T}_3}
$$

$$
\tag{24}
$$

The above Equation can be decomposed into three parts, namely $\mathcal{T}_1$, $\mathcal{T}_2$, and $\mathcal{T}_3$. Notably, each of these parts is associated with weight $\frac{1}{n_i}$. This observation supports Theorem 1 and Lemma 1, which suggest that the variance of a model is highly dependent on the distribution of the dataset's sample size, and that the extent of sample imbalance can significantly increase the model's variance.

Note that $\epsilon^i \sim N(0, \Lambda^i)$, $e^i \sim N \left( 0, \frac{1}{n_i} \Lambda^i \right)$, and so $\frac{1}{\sqrt{n_i}} \epsilon^i \sim N \left( 0, \frac{1}{n_i} \Lambda^i \right)$. So, if we use the class center $C^i$ to replace $h$ in Equation 3, then we can get the following expression,

$$\frac{1}{N} \sum_{x \in G} \sum_{i=1}^{c} \left( (C^i)^T h(x) - (C^{i'})^T h'(x) \right)^2$$

$$= \frac{1}{N} \sum_{k=1}^{n_j} \sum_{j=1}^{c} \sum_{i=1}^{c} \left( (\mu^i)^T (\epsilon_k^j - \epsilon_k^{j'}) + (e^i)^T \epsilon_k^j - (e^{i'})^T \epsilon_k^{j'} \right)^2$$

$$= \frac{1}{N} \sum_{k=1}^{n_j} \sum_{j=1}^{c} \sum_{i=1}^{c} \left( (\mu^i)^T (\epsilon_k^j - \epsilon_k^{j'}) + (\frac{1}{\sqrt{n_i}} \epsilon^i)^T \epsilon_k^j - (\frac{1}{\sqrt{n_i}} \epsilon^i)^T \epsilon_k^{j'} \right)^2$$

$$= \frac{1}{N} \sum_{k=1}^{n_i} \sum_{i=1}^{c} \sum_{j=1}^{c} \left( (\mu^j)^T (\epsilon_k^i - \epsilon_k^{i'}) + (\frac{1}{\sqrt{n_j}} \epsilon^j)^T \epsilon_k^i - (\frac{1}{\sqrt{n_j}} \epsilon^j)^T \epsilon_k^{i'} \right)^2 \qquad (25)$$

$$= \frac{1}{N} \sum_{k=1}^{n_i} \sum_{i=1}^{c} \sum_{j=1}^{c} \left( (\mu^j)^T (\epsilon_k^i - \epsilon_k^{i'}) + (\frac{1}{\sqrt{n_j}} \epsilon^j)^T (\epsilon_k^i - \epsilon_k^{i'}) \right)^2$$

$$= \underbrace{\frac{1}{N} \sum_{k=1}^{n_i} \sum_{i=1}^{c} \sum_{j=1}^{c} \left( \left[ (\mu^j)^T (\epsilon_k^i - \epsilon_k^{i'}) \right]^2 \right)}_{\mathcal{L}_1} + \underbrace{\frac{1}{N} \sum_{k=1}^{n_i} \sum_{i=1}^{c} \sum_{j=1}^{c} \left( \frac{1}{n_j} \left[ (\epsilon^j)^T (\epsilon_k^i - \epsilon_k^{i'}) \right]^2 \right)}_{\mathcal{L}_2}$$

$$+ \underbrace{\frac{2}{N} \sum_{k=1}^{n_i} \sum_{i=1}^{c} \sum_{j=1}^{c} \left( \frac{1}{\sqrt{n_j}} \left[ (\mu^j)^T (\epsilon_k^i - \epsilon_k^{i'}) (\epsilon^j)^T (\epsilon_k^i - \epsilon_k^{i'}) \right] \right)}_{\mathcal{L}_3}$$

Like the previous Equation 24, Equation 25 can be decomposed into three parts: $\mathcal{L}_1$, $\mathcal{L}_2$, and $\mathcal{L}_3$. If we substitute the class center $C^i$ for $h$, we can observe that even though $\mathcal{L}_1$ is insensitive to class imbalance, variables $\mathcal{L}_2$ and $\mathcal{L}_3$, which respectively incorporate weights $\frac{1}{n_j}$ and $\frac{1}{\sqrt{n_j}}$, can still provide the following support: when optimizing the variance of the model, more attention can be given to the variance introduced by the minority classes, which also provides an innovative perspective for understanding class imbalance on graphs. $\qquad \square$

## C   More Results

### C.1   More results Under Traditional Semi-supervised Settings

In Table 3, we present the results of ReVar and other algorithms on Cora-Semi ($\rho$=10). The experimental results demonstrate that TAM has a significant gain effect on all models (BalancedSoftmax, Renode, GraphENS), while GraphENS achieved the best performance among all current baselines on various models. By comparing with GraphSMOTE, we can conclude that addressing the node classification imbalance issue should focus on the topological characteristics of the graph. Finally, and most importantly, ReVar achieved state-of-the-art results under all basic models, which also verifies the superiority of our model in traditional imbalanced segmentation.

Table 3: Experimental results of our method ReVar and other baselines on *Cora-Semi*. We report averaged balanced accuracy (bAcc.,%) and F1-score (%) with the standard errors over 5 repetitions on three representative GNN architectures. Highlighted are the top **first** and **second**. $\Delta$ is the margin by which our method leads state-of-the-art method.

| Dataset(*Cora-Semi*) | GCN | | GAT | | SAGE | |
|---|---|---|---|---|---|---|
| Imbalance Ratio ($\rho = 10$) | bAcc. | F1 | bAcc. | F1 | bAcc. | F1 |
| Vanilla | $62.82 \pm 1.43$ | $61.67 \pm 1.59$ | $62.33 \pm 1.56$ | $61.82 \pm 1.84$ | $61.82 \pm 0.97$ | $60.97 \pm 1.07$ |
| Re-Weight | $65.36 \pm 1.15$ | $64.97 \pm 1.39$ | $66.87 \pm 0.97$ | $66.62 \pm 1.13$ | $63.94 \pm 1.07$ | $63.82 \pm 1.30$ |
| PC Softmax | $68.04 \pm 0.82$ | $67.84 \pm 0.81$ | $66.69 \pm 0.79$ | $66.04 \pm 1.10$ | $65.79 \pm 0.70$ | $66.04 \pm 0.92$ |
| GraphSMOTE | $66.39 \pm 0.56$ | $65.49 \pm 0.93$ | $66.71 \pm 0.32$ | $65.01 \pm 1.21$ | $61.65 \pm 0.34$ | $60.97 \pm 0.98$ |
| BalancedSoftmax | $69.98 \pm 0.58$ | $68.68 \pm 0.55$ | $67.89 \pm 0.36$ | $67.96 \pm 0.41$ | $67.43 \pm 0.61$ | $67.66 \pm 0.69$ |
| + TAM | $69.94 \pm 0.45$ | $69.54 \pm 0.47$ | $69.16 \pm 0.27$ | $69.39 \pm 0.37$ | $69.03 \pm 0.92$ | $69.03 \pm 0.97$ |
| Renode | $67.03 \pm 1.41$ | $67.16 \pm 1.67$ | $67.33 \pm 0.79$ | $68.08 \pm 1.16$ | $66.84 \pm 1.78$ | $67.08 \pm 1.75$ |
| + TAM | $68.26 \pm 1.84$ | $68.11 \pm 1.97$ | $67.50 \pm 0.67$ | $68.06 \pm 0.96$ | $67.28 \pm 1.11$ | $67.15 \pm 1.11$ |
| GraphENS | $70.89 \pm 0.71$ | $70.90 \pm 0.81$ | $70.45 \pm 1.25$ | $69.87 \pm 1.32$ | $68.74 \pm 0.46$ | $68.34 \pm 0.33$ |
| + TAM | $71.69 \pm 0.36$ | $72.14 \pm 0.51$ | $70.15 \pm 0.18$ | $70.00 \pm 0.40$ | $70.45 \pm 0.74$ | $70.40 \pm 0.75$ |
| **ReVar** | $72.92 \pm 2.27$ | $72.60 \pm 2.26$ | $74.56 \pm 0.96$ | $74.61 \pm 0.96$ | $73.32 \pm 3.02$ | $68.91 \pm 3.13$ |
| $\Delta$ | + 1.23(1.72%) | + 0.46(0.64%) | + 4.11(5.83%) | + 4.61(6.59%) | + 2.87(4.07%) | - 1.49(2.12%) |

Table 4: Experimental results of our method ReVar and other baselines on *Computers-Random*. We report averaged balanced accuracy (bAcc.,%) and F1-score (%) with the standard errors over 5 repetitions on three representative GNN architectures. Highlighted are the top **first** and **second**. $\Delta$ is the margin by which our method leads state-of-the-art method.

| Dataset(*Computers-Random*) | GCN | | GAT | | SAGE | |
|---|---|---|---|---|---|---|
| Imbalance Ratio ($\rho = 25.50$) | bAcc. | F1 | bAcc. | F1 | bAcc. | F1 |
| Vanilla | $78.43 \pm 0.41$ | $77.14 \pm 0.39$ | $71.35 \pm 1.18$ | $69.60 \pm 1.11$ | $65.30 \pm 1.07$ | $64.77 \pm 1.19$ |
| Re-Weight | $80.49 \pm 0.44$ | $75.07 \pm 0.26$ | $71.95 \pm 0.80$ | $70.67 \pm 0.51$ | $66.50 \pm 1.47$ | $66.10 \pm 1.46$ |
| PC Softmax | $81.34 \pm 0.55$ | $75.17 \pm 0.57$ | $70.56 \pm 1.46$ | $67.26 \pm 1.48$ | $69.73 \pm 0.53$ | $67.03 \pm 0.6$ |
| GraphSMOTE | $80.50 \pm 1.11$ | $73.79 \pm 0.14$ | $71.98 \pm 0.21$ | $67.98 \pm 0.31$ | $72.69 \pm 0.82$ | $68.73 \pm 1.01$ |
| BalancedSoftmax | $81.39 \pm 0.25$ | $74.54 \pm 0.64$ | $72.09 \pm 0.31$ | $68.38 \pm 0.69$ | $73.80 \pm 1.06$ | $69.74 \pm 0.60$ |
| + TAM | $81.64 \pm 0.48$ | $75.59 \pm 0.83$ | $74.00 \pm 0.77$ | $70.72 \pm 0.50$ | $73.77 \pm 1.26$ | $71.03 \pm 0.69$ |
| Renode | $81.64 \pm 0.34$ | $76.87 \pm 0.32$ | $72.80 \pm 0.94$ | $71.40 \pm 0.97$ | $70.94 \pm 1.50$ | $70.04 \pm 1.16$ |
| + TAM | $80.50 \pm 1.11$ | $75.79 \pm 0.14$ | $71.98 \pm 0.21$ | $70.98 \pm 0.31$ | $72.69 \pm 0.82$ | $70.73 \pm 1.01$ |
| GraphENS | $82.66 \pm 0.61$ | $76.55 \pm 0.17$ | $75.25 \pm 0.85$ | $71.49 \pm 0.54$ | $77.64 \pm 0.52$ | $72.65 \pm 0.53$ |
| + TAM | $82.83 \pm 0.68$ | $76.76 \pm 0.39$ | $75.81 \pm 0.72$ | $72.62 \pm 0.57$ | $78.98 \pm 0.60$ | $73.59 \pm 0.55$ |
| **ReVar** | $85.00 \pm 0.07$ | $82.35 \pm 0.08$ | $81.94 \pm 0.54$ | $80.94 \pm 0.25$ | $80.61 \pm 0.11$ | $77.49 \pm 0.09$ |
| $\Delta$ | + 2.17(2.62%) | + 5.48(7.13%) | + 6.13(8.09%) | + 8.32(11.46%) | + 1.63(2.06%) | + 3.90(5.30%) |

### C.2   More results On Naturally Imbalanced Datasets

One of our major highlights is testing our algorithms on naturally imbalanced datasets, which is more representative of real-world scenarios. This means that in a semi-supervised setting, our training set and unlabeled data are both imbalanced, which poses a significant challenge for oversampling methods. In Table 4, we present the results of ReVar and all baselines on *Computers-Random*,

demonstrating that our model still achieves state-of-the-art performance. In contrast, oversampling methods such as GraphSMOTE perform poorly on this dataset due to overfitting limitations. We believe that our model's ability to regularize variance from a bottom-up perspective contributes to its superior performance on naturally imbalanced graphs, while also demonstrating its powerful generalization capabilities.

# D More Analysis

## D.1 More Ablation Analysis for the Loss Function

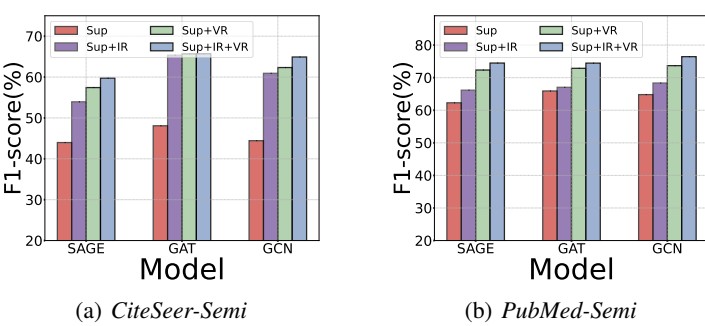

(a) *CiteSeer-Semi*          (b) *PubMed-Semi*

Figure 4: More Ablation Analysis for the Loss Function.

**Details of Experimental Setup.** We conduct more experiments on imbalance ratio 10 ($\rho = 10$) for the datasets CiteSeer and PubMed. For each GNN network, we add one loss regularization at a time, i.e. for GCN, we gradually add $\mathcal{L}_{\text{IR}}$, $\mathcal{L}_{\text{VR}}$ from the initial $\mathcal{L}_{\text{sup}}$. The architecture we employed consisted of a 2-layers graph neural network (GNN) with 128 hidden dimensions, using GCN [16], GAT [35], and GraphSAGE [9]. The models were trained for 2000 epochs.

**Analysis.** Figure 4 reports more experiments for each component in the loss function. As we have seen, each component of the loss function can bring an improvement in the training effect. It is worth noting that in most cases $\mathcal{L}_{\text{VR}}$ has a larger effect boost than $\mathcal{L}_{\text{IR}}$. We believe that $\mathcal{L}_{\text{VR}}$ plays a more important role in the loss function.

## D.2 More Experiments for Variance and Imbalance Ratio Correlation in Theorem 1

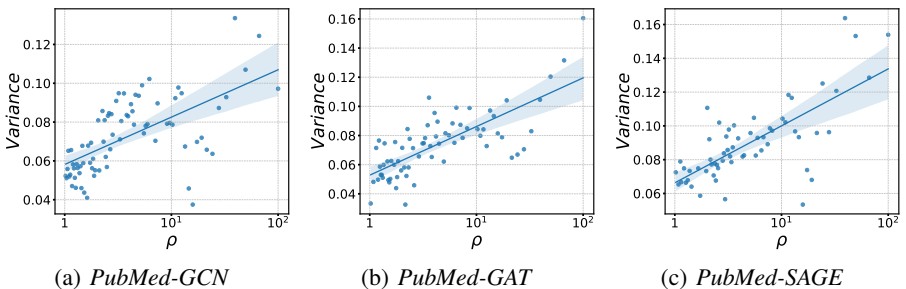

(a) *PubMed-GCN*          (b) *PubMed-GAT*          (c) *PubMed-SAGE*

Figure 5: More experiments for variance and imbalance ratio correlation in Theorem 1.

**Details of Experimental Setup.** In this experiment, the classifier is not MLP, which means that the probability of a node $v$ being classified into class $i$ depends on the distance of this node from the center of class $i$. We classify half of the classes as majority classes and the other half as minority classes. Initially, each class in majority classes and minority classes has 200 labeled nodes. To generate different imbalance scenarios ($\rho$), we reduce the number of labeled nodes in the minority class by 1 and add 1 to the number of labeled nodes in the majority class each time, so as to keep the number of samples in the training set consistent and eliminate the effect of the sample size variance in the training set. To calculate the variance of the model under a certain imbalance ratio ($\rho$), we repeat 20 times to randomly select different but the same number of training sets, and train 2000 epochs for each fixed training set. The architecture we employed consisted of a 2-layers graph neural network (GNN) with 128 hidden dimensions, using GCN [16], GAT [35], and GraphSAGE [9].

**Hypothesis testing.** To test the correlation between the variance and imbalance ratio, we computed the Pearson correlation coefficient ($\rho$) between variance and imbalance ratio (log), as presented in the table below. The Pearson correlation coefficient, denoted as $r$, is a prevalent metric for gauging linear correlations. This coefficient lies between (-1) and (1) ,and reflects both the magnitude and direction of the correlation between two variables. An ($r$) value greater than (0.5) indicates a strong positive correlation. Furthermore, the p-value results from a hypothesis test with the null hypothesis ($H_0 : \rho = 0$) and the alternative hypothesis ($H_1 : \rho \neq 0$) , where represents the population correlation coefficient.

|   | Citeseer-GCN | Citeseer-GAT | Citeseer-SAGE | PubMed-GCN | PubMed-GAT | PubMed-SAGE |
|---|---|---|---|---|---|---|
| **r** | 0.751 | 0.786 | 0.642 | 0.694 | 0.760 | 0.747 |
| **p-value** | $3.203{\times}10^{-14}$ | $1.516{\times}10^{-14}$ | $5.107{\times}10^{-07}$ | $2.233{\times}10^{-09}$ | $2.344{\times}10^{-14}$ | $5.634{\times}10^{-13}$ |

Given that the Pearson correlation coefficient between variance and imbalance ratio exceeds (0.5), and the p-value is below (0.01), we deduce that there is a robust correlation between variance and imbalance ratio. This relationship is statistically significant at the (0.01) significance level.

**Analysis.** Figure 5 reveals an intriguing pattern, as the majority of data points closely align along a regression curve exhibiting a positive slope. This observation provides substantial evidence that establishes a strong and direct linear relationship between the imbalance ratio ($\rho$) and the associated variance. Consequently, our findings provide compelling support for the hypothesis postulated in Theorem 1.

### D.3 More Experiments for Hyperparameter Sensitivity Analysis of ReVar

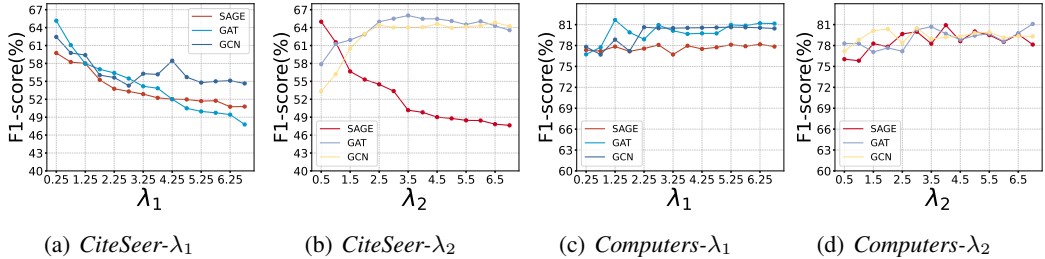

(a) *CiteSeer-$\lambda_1$*  (b) *CiteSeer-$\lambda_2$*  (c) *Computers-$\lambda_1$*  (d) *Computers-$\lambda_2$*

Figure 6: More Experiments of Hyperparameter Sensitivity Analysis for ReVar.

**Details of Experimental Setup.** We implemented experiments on the two datasets, CiteSeer and Amazon-Computers. In this experiment, we evaluate the F1 score when we fix one $\lambda$ while the other changes. The architecture we employed consisted of a 2-layer graph neural network (GNN) with 128 hidden dimensions, using GCN [16], GAT [35], and GraphSAGE [9]. The models were trained for 2000 epochs.

**Analysis.** In Figure 6, we present the sensitivity analysis of ReVar to two weight hyperparameters ($\lambda_1$, $\lambda_2$) in the loss function on the CiteSeer and Amazon-Computers. It is evident that ReVar exhibits varying degrees of sensitivity to hyperparameters on different datasets. Specifically, the model demonstrates lower sensitivity to a and b on CiteSeer, while the opposite is observed on Amazon-Computers. We postulate that the number of nodes (i.e., dataset size) may be a crucial factor in this discrepancy. Nevertheless, the optimal range of hyperparameter selection appears to be relatively narrow, indicating that significant efforts are not be required for model fine-tuning.

### D.4 More Results of loss curve and F1 score

**Details of Experimental Setup.** We implemented experiments on three datasets PubMed, CiteSeer and Amazon-Computers. We compare our method ReVar and other vanilla models (only using cross-entropy and not using graph augmentation). The architecture we employed for ReVar and

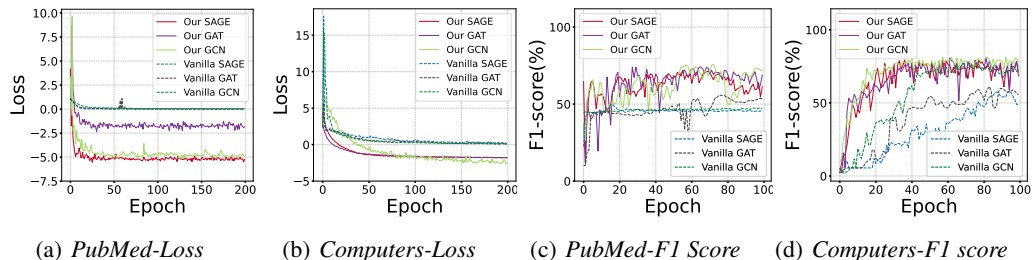

| (a) *PubMed-Loss* | (b) *Computers-Loss* | (c) *PubMed-F1 Score* | (d) *Computers-F1 score* |

Figure 7: More analysis of loss curve and F1 score.

vanilla model consisted of a 2-layer graph neural network (GNN) with 128 hidden dimensions, using GCN [16], GAT [35], and GraphSAGE [9]. The models were trained for 2000 epochs.

**Analysis.** We plot the loss curve and F1 score with epoch on the PubMed and Computers datasets. By analyzing the plotted loss curves, we observe that our model demonstrates notable advantages, including faster convergence of the loss curves and improved performance across multiple datasets. For the F1 score, our model consistently outperforms alternative approaches, showcasing its superior capability in effectively capturing the complex relationships within the data and making accurate predictions.

# E Elaboration of the experimental setup

In this section, we present our approach for constructing imbalanced datasets, describe our evaluation protocol, and provide comprehensive details on our algorithm as well as the baseline methods utilized in our study. To achieve this, we leverage sophisticated techniques and utilize advanced metrics to ensure the reliability and relevance of our results.

## E.1 Construction for Imbalanced datasets

The detailed descriptions of the datasets are shown in Table 5. The details of label distribution in the training set of the five imbalanced benchmark datasets are in Table 7, and the label distribution of the full graph is provided in Table 7.

Table 5: Summary of the datasets used in this work.

| Dataset | Nodes | Edges | Features | Classes |
|---|---|---|---|---|
| *Cora* | 2,708 | 5,429 | 1,433 | 7 |
| *Citeseer* | 3,327 | 4,732 | 3,703 | 6 |
| *Pubmed* | 19,717 | 44,338 | 500 | 3 |
| *Amazon-Computers* | 13,752 | 491,722 | 767 | 10 |
| *Coauthor-CS* | 18,333 | 163,788 | 6,805 | 15 |

**Imbalanced datasets construction for Traditional Semi-supervised Settings.** For each citation dataset, we adopt the "public" split and apply a random undersampling technique to make the class distribution imbalanced until the target imbalance ratio $\rho$ is achieved. Specifically, we convert the minority class nodes to unlabeled nodes in a random manner. Regarding the co-purchased networks Amazon-Computers, we conduct replicated experiments by randomly selecting nodes as the training set in each trial. We create a random validation set that contains 30 nodes in each class, and the remaining nodes are used as the testing set.

**Imbalanced datasets construction for Naturally Imbalanced Datasets.** For *Computers-Random* and *CS-Random*, we constructed a training set with equal proportions based on the label distribution of the complete graph (Amazon-Computers). The label distributions for the training sets of *Computers-Random* and *CS-Random* are presented in Table 7. To achieve this, we employed a stratified sampling approach that ensured an unbiased representation of all labels in the training set.

Table 6: Label distributions in the training sets

| Dataset | $\mathcal{C}_0$ | $\mathcal{C}_1$ | $\mathcal{C}_2$ | $\mathcal{C}_3$ | $\mathcal{C}_4$ | $\mathcal{C}_5$ | $\mathcal{C}_6$ | $\mathcal{C}_7$ | $\mathcal{C}_8$ | $\mathcal{C}_9$ | $\mathcal{C}_{10}$ | $\mathcal{C}_{11}$ | $\mathcal{C}_{12}$ | $\mathcal{C}_{13}$ | $\mathcal{C}_{14}$ |
|---|---|---|---|---|---|---|---|---|---|---|---|---|---|---|---|
| *Cora-Semi* ($\rho = 10$) | 23.26% (20) | 23.26% (20) | 23.26% (20) | 23.26% (20) | 2.32% (2) | 2.32% (2) | 2.32% (2) | - | - | - | - | - | - | - | - |
| *CiteSeer-Semi* ($\rho = 10$) | 30.30% (20) | 30.30% (20) | 30.30% (20) | 3.03% (2) | 3.03% (2) | 3.03% (2) | - | - | - | - | - | - | - | - | - |
| *PubMed-Semi* ($\rho = 10$) | 47.62% (20) | 47.62% (20) | 4.76% (2) | - | - | - | - | - | - | - | - | - | - | - | - |
| *Computers-Semi* ($\rho = 10$) | 18.18% (20) | 18.18% (20) | 18.18% (20) | 18.18% (20) | 18.18% (20) | 1.82% (2) | 1.82% (2) | 1.82% (2) | 1.82% (2) | 1.82% (2) | - | - | - | - | - |
| *Computers-Random* ($\rho = 25.50$) | 3.01% (4) | 15.79% (21) | 10.53% (14) | 3.76% (5) | 38.35% (51) | 2.26% (3) | 3.01% (4) | 6.02% (8) | 15.79% (21) | 1.50% (2) | - | - | - | - | - |
| *CS-Random* ($\rho = 41.00$) | 3.98% (7) | 2.27% (4) | 11.36% (20) | 2.27% (4) | 7.39% (13) | 11.93% (21) | 1.70% (3) | 5.11% (9) | 3.98% (7) | 0.57% (1) | 7.95% (14) | 11.36% (20) | 2.27% (4) | 23.30% (41) | 4.55% (8) |

Table 7: Label distributions on the whole graphs

| Dataset | $\mathcal{C}_0$ | $\mathcal{C}_1$ | $\mathcal{C}_2$ | $\mathcal{C}_3$ | $\mathcal{C}_4$ | $\mathcal{C}_5$ | $\mathcal{C}_6$ | $\mathcal{C}_7$ | $\mathcal{C}_8$ | $\mathcal{C}_9$ | $\mathcal{C}_{10}$ | $\mathcal{C}_{11}$ | $\mathcal{C}_{12}$ | $\mathcal{C}_{13}$ | $\mathcal{C}_{14}$ |
|---|---|---|---|---|---|---|---|---|---|---|---|---|---|---|---|
| *Cora* ($\rho \approx 4.54$) | 351 | 217 | 418 | 818 | 426 | 298 | 180 | - | - | - | - | - | - | - | - |
| *CiteSeer* ($\rho \approx 2.66$) | 264 | 590 | 668 | 701 | 696 | 508 | - | - | - | - | - | - | - | - | - |
| *PubMed* ($\rho \approx 1.91$) | 4103 | 7739 | 7835 | - | - | - | - | - | - | - | - | - | - | - | - |
| *Amazon-Computers* ($\rho \approx 17.73$) | 436 | 2142 | 1414 | 542 | 5158 | 308 | 487 | 818 | 2156 | 291 | - | - | - | - | - |
| *Coauthor-CS* ($\rho \approx 35.05$) | 708 | 462 | 2050 | 429 | 1394 | 2193 | 371 | 924 | 775 | 118 | 1444 | 2033 | 420 | 4136 | 876 |

## E.2 Architecture of GNNs

We conducted evaluations using three classic GNN architectures, namely GCN [16], GAT [35], and GraphSAGE [9]. The GNN models were constructed with different numbers of layers, namely $L = 1, 2, 3$. To enhance the learning process, each GNN layer was accompanied by a BatchNorm layer with a momentum of 0.99, followed by a PRelu activation function [11]. For the GAT architecture, we employed multi-head attention with 8 heads. We search for the best model on the validation set. The available choices for the hidden unit sizes were 64, 128, and 256.

### E.3   Evaluation Protocol

We utilized the Adam optimizer [15] with an initial learning rate of 0.01 or 0.005. To manage the learning rate, we employed a scheduler based on the approach outlined in [31], which reduced the learning rate by half when there was no decrease in validation loss for 100 consecutive epochs. Weight decay with a rate of 0.0005 was applied to all learnable parameters in the model. In the initial training iteration, we trained the model for 200 epochs using the original training set for Cora, CiteSeer, PubMed, or Amazon-Computers. However, for Flickr, the training was extended to 2000 epochs in the first iteration. Subsequently, in the remaining iterations, we trained the models for 2000 epochs using the aforementioned optimizer and scheduler. The best models were selected based on validation accuracy, and we employed early stopping with a patience of 300 epochs.

### E.4   Technical Details of ReVar

For all datasets *Cora-Semi*, *CiteSeer-Semi*, *PubMed-Semi*, *Computers-Semi*, *Computers-Random* and *CS-Random*, the learning rate $\eta$ is chosen from $\{0.0001, 0.0005, 0.001, 0.005, 0.01, 0.1\}$. The temperature hyperparameter $\tau$ is chosen from $\{0.05, 0.08, 0.13, 0.16, 0.21, 0.23, 0.26\}$. The threshold $v$ is chosen from $\{0.6, 0.63, 0.66, 0.7, 0.8, 0.83, 0.9, 0.93, 0.96, 0.99\}$. The factor $\lambda_1$ of $\mathcal{L}_{\text{VR}}$ is chosen from $\{0.25, 0.35, 0.5, 0.85, 1, 1.5, 2, 2.15, 2.65, 3\}$. The factor $\lambda_2$ of $\mathcal{L}_{\text{IR}}$ is chosen from $\{0.35, 0.5, 1, 1.25, 1.5, 2.85, 3\}$. The factors of mask node properties and edges in $\tilde{G}$ are chosen from $\{0.4, 0.45, 0.5, 0.6, 0.65, 0.7\}$ and $\{0.4, 0.45, 0.5, 0.55, 0.6, 0.65, 0.7\}$. The factors of mask node properties and edges in $\tilde{G}'$ are chosen from $\{0.1, 0.15, 0.2, 0.3, 0.4, 0.45\}$ and $\{0.1, 0.15, 0.2, 0.3, 0.35, 0.4, 0.45\}$.

### E.5   Technical Details of Baselines

For the GraphSMOTE method [50], we employed the branched algorithms whose edge predictions are discrete-valued, which have achieved superior performance over other variants in most experiments. Regarding the ReNode method [6], we search hyperparameters within the lower bound of cosine annealing, $w_{\min} \in 0.25, 0.5, 0.75$, and the upper bound range, $w_{\max} \in 1.25, 1.5, 1.75$, as suggested in [6]. The PageRank teleport probability was fixed at $\alpha = 0.15$, which is the default setting in the released codes. As for TAM [31], we performed a hyperparameter search for the coefficient of the ACM term, $\alpha \in 1.25, 1.5, 1.75$, the coefficient of the ADM term, $\beta \in 0.125, 0.25, 0.5$, and the minimum temperature of the class-wise temperature, $\phi \in 0.8, 1.2$, following the approach described in [31]. The sensitivity to the imbalance ratio of the class-wise temperature, $\delta$, was fixed at 0.4 for all the main experiments. Additionally, consistent with [31], we implemented a warmup phase for 5 iterations, as we utilized model predictions for unlabeled nodes.

### E.6   Configuration

All the algorithms and models are implemented in Python and PyTorch Geometric. Experiments are conducted on a server with an NVIDIA 3090 GPU (24 GB memory) and an Intel(R) Xeon(R) Silver 4210R CPU @ 2.40GHz.

# F  Algorithm

---

**Algorithm 1** *ReVar*

---

**Input:** Imbalanced Graph $\mathbf{G} = (\mathbf{V}, \mathbf{E}, \mathbf{V_L})$, feature matrix $\mathbf{X}$, adjacency matrix $\mathbf{A}$, unlabeled set $\mathbf{V_U} = \mathbf{V} - \mathbf{V_L}$, label matrix $\mathbf{Y}$, feature matrix of labeled nodes and unlabeled nodes $\mathbf{X_L}$ and $\mathbf{X_U}$, the number of classes $\mathbf{k}$, the threshold $v$ for determining whether a node has confident prediction, temperature hyperparameter $\tau$, GNN model $f_\theta$, the classifier $h_\theta$, learning rate $\eta$, The total number $T$ of epochs for model training. The $\mathrm{sim}(\cdot, \cdot)$ computes the cosine similarity between two vectors, *CE* represents the cross-entropy loss function, and $\mathbb{I}(\cdot)$ is an indicator function.

1:  **for** $t = 0, 1, \ldots, T$ **do**
2:      Generate two differently augmented views $\tilde{G} = (\tilde{\mathrm{A}}, \tilde{\mathrm{X}})$ and $\tilde{G}' = (\tilde{\mathrm{A}}', \tilde{\mathrm{X}}')$ from original graph $\mathbf{G}$.
3:      $\tilde{\mathbf{O}}_L \leftarrow f_\theta(\tilde{\mathrm{A}}, \tilde{\mathrm{X}}_L), \tilde{\mathbf{O}}_U \leftarrow f_\theta(\tilde{\mathrm{A}}, \tilde{\mathrm{X}}_U)$
4:      $\tilde{\mathbf{O}}'_L \leftarrow f_\theta(\tilde{\mathrm{A}}', \tilde{\mathrm{X}}'_L), \tilde{\mathbf{O}}'_U \leftarrow f_\theta(\tilde{\mathrm{A}}', \tilde{\mathrm{X}}'_U)$
5:      **% Component 1: Supervised Loss**
6:      $\mathcal{L}_{sup} \leftarrow \frac{1}{2} CE(h_\theta(\tilde{\mathbf{O}}_L), \mathrm{Y}) + \frac{1}{2} CE(h_\theta(\tilde{\mathbf{O}}'_U), \mathrm{Y})$
7:      **% Component 2: Intra-Class Aggregation Regularization**
8:      $\mathcal{L}_{\mathrm{IR}} \leftarrow -\frac{1}{N_U} \sum_{i=1}^{N_U} \mathrm{sim}(\tilde{\mathbf{O}}_i, \tilde{\mathbf{O}}'_i) - \frac{1}{N_{all}} (\sum_{i=1}^{N_L} \sum_{\substack{j=1 \\ (i,j) \in same}}^{N_L} \mathrm{sim}(\tilde{\mathbf{O}}_i, \tilde{\mathbf{O}}'_j) + \sum_{i=1}^{N_L} \sum_{\substack{j=1, j \neq i \\ (i,j) \in same}}^{N_L} \mathrm{sim}(\tilde{\mathbf{O}}_i, \tilde{\mathbf{O}}_j)$
9:      **% Component 3: Variance-constrained Optimization with Adaptive Regularization**
10:     **for** i = 1, 2, $\ldots$, $k$ **do**
11:         Compute the class centers $\tilde{\mathbf{C}}_i$ and $\tilde{\mathbf{C}}'_i$ for class $i$
12:     **end for**
13:     $\triangleright$ Obtain the Label Probability for Each Node
14:     **for** $i = 1, 2, \ldots, |\mathbf{V}|$ **do**
15:         $\tilde{\mathbf{p}}_i \leftarrow \mathrm{Softmax}(\tilde{\mathbf{O}}_i \cdot \left[ \tilde{\mathbf{C}}_1, \ldots, \tilde{\mathbf{C}}_k \right]^T)$
16:         $\tilde{\mathbf{p}}'_i \leftarrow \mathrm{Softmax}\left( \tilde{\mathbf{O}}'_i \cdot [\tilde{\mathbf{C}}'_1, \ldots, \tilde{\mathbf{C}}'_k]^T \right)$
17:     **end for**
18:     $\triangleright$ Eliminate Nodes with Low Confidence.
19:     $\mathbf{V}_{\mathrm{conf}} \leftarrow \{i \mid \mathbb{I}(\max(\tilde{\mathbf{p}}'_i) > v) = 1, \forall i \in V_U\}$
20:     $\triangleright$ Replace Labeled Node Prediction
21:     **for** $i = 1, 2, \ldots, |\mathbf{V_L}|$ **do**
22:         $\tilde{\mathbf{p}}'_i \leftarrow \mathbf{Y}_i$
23:     **end for**
24:     $\mathcal{L}_{\mathbf{VR}} \leftarrow \frac{1}{|V_{\mathbf{conf}}|} \sum_{i \in V_{\mathbf{conf}}} CE(\tilde{\mathbf{p}}'_i, \tilde{\mathbf{p}}_i) + \frac{1}{|V_L|} \sum_{v_i \in V_L} CE(\mathbf{Y}_i, \tilde{\mathbf{p}}_i)$
25:     $\mathcal{L}_{Training} \leftarrow \mathcal{L}_{sup} + \lambda_1 \mathcal{L}_{VR} + \lambda_2 \mathcal{L}_{IR}$
26:     $\theta \leftarrow \theta - \eta (\nabla \mathcal{L}_{Training})$
27: **end for**

---

# G   Notations

Table 8: Notation table.

| *Indices* | |
| --- | --- |
| $n$ | The number of nodes, $n = |\mathbf{V}|$. |
| $f$ | The node feature dimension. |
| $k$ | The number of different classes. |
| $D$ | The dimension of the embedding space, or the dimension of the last layer of GNNs. |
| *Parameters* | |
| $\mathbf{G}$ | An undirected and unweighted graph. |
| $\mathbf{V}$ | The node set of $\mathbf{G}$. |
| $\mathbf{E}$ | The edge set of $\mathbf{G}$. |
| $\mathbf{X}$ | The feature matrix of $\mathbf{G}$, $\mathbf{X} \in \mathbb{R}^{n \times f}$. |
| $\mathbf{V_L}$ | The set of labeled nodes of $\mathbf{G}$. |
| $\mathbf{A}$ | The adjacency matrix of $\mathbf{G}$, $\mathbf{A} \in \{0,1\}^{n \times n}$. |
| $\mathbf{N}(v)$ | The set of 1-hop neighbors for node $v$. |
| $\mathbf{V_U}$ | The set of unlabeled nodes, $\mathbf{V_U} := \mathbf{V} \setminus \mathbf{V_L}$. |
| $\mathbf{C}_i$ | The labeled set for class $i$. |
| $\rho$ | Imbalance ratio of a dataset, $\rho = \max_i |\mathbf{C}_i| / \min_i |\mathbf{C}_i|$. |
| $\tilde{\mathbf{G}}$ | A view from the original graph $\mathbf{G}$ , $\tilde{\mathbf{G}} = \left( \tilde{\mathbf{A}}, \tilde{\mathbf{X}} \right)$ . |
| $\tilde{\mathbf{G}}'$ | Another view from the original graph $\mathbf{G}$ , $\tilde{\mathbf{G}}' = \left( \tilde{\mathbf{A}}', \tilde{\mathbf{X}}' \right)$. |
| h | Output of GNN network, $\mathrm{h} = f_\theta(\mathbf{A}, \mathbf{X})$. |
| $\mathcal{S}$ | The probability distribution reference matrix, $\mathcal{S} := [C_1; C_2; \dots ; C_k]$. |
| $C$ | The collection of class centers, $C := (C_1; C_2; \dots ; C_k)$. |
| $\pi_i$ | The probability distribution for node $i$. |
| $y_i$ | The one-hot label vector for node $i$. |
| $\mathcal{L}_{\text{sup}}$ | Supervised loss. |
| $\mathcal{L}_{\text{IR}}$ | Intra-class aggregation regularization. |
| $\mathcal{L}_{\text{VR}}$ | Variance-constrained optimization with adaptive regularization. |
| $\mathcal{L}_{\text{final}}$ | The sum of all losses. |
| $V_{\text{conf}}$ | The set of nodes with confident predictions. |
| $\lambda_1$ | The weight for $\mathcal{L}_{\text{VR}}$. |
| $\lambda_2$ | The weight for $\mathcal{L}_{\text{IR}}$. |
| $v$ | The threshold for determining whether a node has confident prediction. |
| $\tau$ | The temperature hyperparameter. |
| $\alpha$ | The size threshold of nodes being added in each class per round. |
| $\eta$ | Learning rate of GNN model. |
| *Functions* | |
| $\mathbb{I}(\cdot)$ | The indicater function. |
| $\text{sim}(\cdot)$ | The function that computes the cosine similarity between two vectors. |
| $CE(\cdot)$ | The cross-entropy function. |
| $f_\theta$ | GNN model. |

