# OpenReview forum: "Rethinking Semi-Supervised Imbalanced Node Classification from Bias-Variance Decomposition"
_NeurIPS.cc/2023/Conference — NeurIPS 2023 poster_

### Official Review · Reviewer_8yVE · 2023-06-30

**Soundness:** 3 good
**Presentation:** 3 good
**Contribution:** 3 good
**Rating:** 6
**Confidence:** 1

**Summary:**

The paper studies imbalanced node classification problem and propose  a novel perspective to understand graph imbalance via bias-variance decomposition.By leveraging graph data augmentation, the paper develops a regularization technique to approximate the model's variance. The effectiveness of the method is evaluated across multiple settings and datasets, where the proposed method largely outperforms the compared baselines.

**Strengths:**

- The idea is novel and has solid theoretical motivation.
- Strong performance improvement + extensive experiments.
- The paper is clearly written and well-structured.

**Weaknesses:**

- As someone not working directly in the same field, I think it would be very helpful if the related work section could be more comprehensive than its current shape. Plus, the method design in Sec. 4 shares similarity with many existing techniques. For example, the "confidence-based label-guided consistency regularization" in Eq. 8 has been widely used in standard semi-supervised learning (UDA, FixMatch, FlexMatch) for a long time. And the idea of intra-class aggregation for contrastive learning is also studied in Supervised Contrastive Learning paper. IMHO, it would be better if the paper could discuss the difference with them and tune down accordingly, which does not harm the originality of this paper.

Unconfident comments:
- I appreciate the analysis on that overall variance increases with the increase of imbalance ratio. But it is a bit surprising there was not any analysis like this before even in the field of long-tailed recognition. Could the authors please comment on this?

**Questions:**

- What value does "v" at Line 208 take? Is it a fixed value across the training or it is dynamically adapted?
- What does it mean in Sec. 2.2 that "We randomly mask node properties"? Does it mean dropping nodes and the corresponding edges of the original graph at random? Is this a standard way of performing data augmentation on graph? What are other alternative data augmentation methods? It would be great if the authors could give more information on this either in Sec. 2.2 or in related work.

---

> ### Author Rebuttal · Authors · 2023-08-08
>
> We appreciate your comments that the theoretical and experimental contribution of RVGNN is strong for tackling imbalance problems. We address all your concerns below:
> ***
> **Q1**:The related work section could be more detailed. The method in Sec. 4 resembles existing techniques. Discussing differences and making adjustments won't affect the paper's originality.
>
> **Answer1**:
> We appreciate your recognition of the similarities between our method and existing techniques [1,2,3], and we agree that a more detailed comparison with these techniques would enhance the paper.  we would like to kindly draw your attention to **Appendix A where we have provided more detailed related work** including Imbalanced Learning in the Vision Domain and Graph Contrastive Learning. Here is an attempt to address your concern.
> - **Related work of our confidence-based label-guided consistency regularization**:  You are right in pointing out that the method in Eq. 8 shares similarities with widely-used semi-supervised learning techniques like UDA, FixMatch, and FlexMatch. Both UDA[1] and FixMatch[2] emphasize consistent labels after enhanced data augmentation. Consistency loss is skipped for low-confidence samples. FlexMatch[3] challenges a fixed threshold, advocating for adaptive adjustment of pseudo-label status via curriculum learning during training.
> - **We distinguish our work from them as follows**:
>   - First, the confidence-based label-guided consistency regularization we use is **to better estimate the variance of the model** accurately and reduce the interference of samples whose samples are poorly predicted.
>   -  **Our approach considers the underlying graph structure, exploiting the relationships between nodes to guide the consistency regularization process. This adaptation leads to more coherent propagation of information across graph structures, which distinguishes our method from the conventional techniques mentioned**. However, we acknowledge the importance of making these distinctions more explicit and will revise the section to articulate these differences more clearly.
> - **Related work of our Intra-class Aggregation for Contrastive Learning**: we also acknowledge the similarities between our method and the ideas studied in the Supervised Contrastive Learning[4] paper. Our intra-class aggregation is indeed conceptually similar, but we employ this concept differently to suit the particularities of graph-structured data.
>    - By considering the relational information in graphs and employing an innovative aggregation strategy, our method strives to capture deeper inter-node dependencies, which differentiates it from the approach taken in Supervised Contrastive Learning. We will provide a more explicit comparison between these methods in the revised manuscript to clarify the novel aspects of our approach. **Importantly, within the GNN domain, we are the first to introduce the idea of supervised contrastive learning into imbalance node classification, which can inspire follow-up work, and we think this is one of our contributions.**
>
> **We agree with your suggestion that discussing these differences and drawing connections with existing methods would not harm the originality of our paper.**
>
> [1] Xie, Qizhe, et al. "Unsupervised data augmentation for consistency training." Advances in neural information processing systems 33 (2020): 6256-6268.
>
> [2] Sohn, Kihyuk, et al. "Fixmatch: Simplifying semi-supervised learning with consistency and confidence." Advances in neural information processing systems 33 (2020): 596-608.
>
> [3] Zhang, Bowen, et al. "Flexmatch: Boosting semi-supervised learning with curriculum pseudo labeling." Advances in Neural Information Processing Systems 34 (2021): 18408-18419.
>
> [4] Khosla, Prannay, et al. "Supervised contrastive learning." Advances in neural information processing systems 33 (2020): 18661-18673.
> ***
> **Q2**: The author values the variance analysis with imbalance ratio but is surprised it's absent in long-tailed recognition.
>
> **Answer2**:
> - We sincerely value the reviewer's keen insights and feedback on this matter. Upon conducting an exhaustive literature review, it appears that **our study is among the pioneering works to investigate the relationship between variance and imbalance ratio**. We also believe that our focus on optimizing the model's variance through theoretical analysis in both graph learning and traditional imbalanced learning is novel. Recognizing the importance of this contribution, we have been meticulous in our theoretical approach. As detailed in Section 6, we have plans to extend RVGNN and its foundational theories to broader areas, including computer vision and natural language processing.
> - To further clarify, we'd like to emphasize that our work currently seems best suited for graph-specific settings. In optimizing the approximate variance, graph data augmentation(GDA) is crucial as it serves to emulate diverse training sets. In this work, it's worth noting that GDA is distinct from traditional data augmentation. Here, GDA acts as a form of dataset augmentation.  It not only perturbs the features of individual nodes but also significantly modifies the graph dataset by cropping or adding edges. However, we recognize that applying our framework to traditional settings may not be direct.
> ***
> **Q3**: Value of "v" at Line 8.
>
> **Answer3**: The value  "v" serves as a threshold to determine whether a node's prediction is confident. It is treated as a hyperparameter.
>
> ***
> **Q4**: The meaning of "randomly mask node properties" and other alternative graph augmentation methods.
>
> **Answer4**:
> - Randomly masking node properties involves randomly **replacing or hiding specific attributes of certain nodes** within the graph.
> - Other augmentation methods include node/edge perturbation, subgraph sampling, feature noise addition, and so on.
>
> We will expand on these points in **detail in the related work section** to provide a more comprehensive explanation.

---

> > ### Author Response · Authors · 2023-08-17
> > **Further Clarification on Q4 of Reviewer 8yVE.**
> >
> > ## 1. Explanation of Randomly Masking Node Properties
> > - "Randomly masking node properties" [1,2,3,4,5] does not refer to randomly dropping nodes and the corresponding edges of the original graph. Instead, it's a data preprocessing technique that involves randomly replacing or hiding specific attributes of certain nodes within the graph. This process is typically used during the training phase and helps the model learn to extract useful information from incomplete or partially corrupted data.
> >
> > [1] You, Yuning, et al. "Graph contrastive learning with augmentations." Advances in neural information processing systems 33 (2020): 5812-5823.
> >
> > [2] Zhao, Tong, et al. "Data augmentation for graph neural networks." Proceedings of the aaai conference on artificial intelligence. Vol. 35. No. 12. 2021.
> >
> > [3] Zhao, Tong, et al. "Graph data augmentation for graph machine learning: A survey." arXiv preprint arXiv:2202.08871 (2022).
> >
> > [4] Ding, Kaize, et al. "Data augmentation for deep graph learning: A survey." ACM SIGKDD Explorations Newsletter 24.2 (2022): 61-77.
> >
> > [5] Zhou, Jiajun, Jie Shen, and Qi Xuan. "Data augmentation for graph classification." Proceedings of the 29th ACM International Conference on Information & Knowledge Management. 2020.
> >
> > ***
> > ## 2. Is this a Standard Way of Performing Data Augmentation on Graph?
> > - Randomly masking node properties can be considered a data augmentation method, particularly in the training of Graph Neural Networks (GNNs). By presenting the model with some masked or perturbed node information, it can enhance the robustness and generalization ability of the model.
> >
> > ***
> > ## 3. What are Other Alternative Data Augmentation Methods?
> > In addition to randomly masking node properties, there are many other methods [1,2,3,4,5,6,7,8] for data augmentation in graphs, including but not limited to:
> > - Node/Edge Perturbation: Altering the graph's structure by randomly adding or removing some nodes or edges.
> > - Subgraph Sampling: Randomly sampling subgraphs from the original graph for use as training samples.
> > - Feature Noise Addition: Adding random noise to the features of nodes or edges.
> > - Graph Rotation and Reflection:  Applying geometric transformations like rotation and reflection to the graph structure.
> > - Adjacency Matrix Perturbation:  Changing the connectivity of the graph by altering the weights of the adjacency matrix.
> >
> > These methods can be used independently or in combination to enrich the training data and enhance the model's robustness and generalization performance. **We will expand on these points in detail in either Sec. 2.2 or in the related work section, to provide a more comprehensive background and explanation.** Thank you again for your valuable feedback!
> >
> > [1] You, Yuning, et al. "Graph contrastive learning with augmentations." Advances in neural information processing systems 33 (2020): 5812-5823.
> >
> > [2] Zhao, Tong, et al. "Data augmentation for graph neural networks." Proceedings of the aaai conference on artificial intelligence. Vol. 35. No. 12. 2021.
> >
> > [3] Zhao, Tong, et al. "Graph data augmentation for graph machine learning: A survey." arXiv preprint arXiv:2202.08871 (2022).
> >
> > [4] Ding, Kaize, et al. "Data augmentation for deep graph learning: A survey." ACM SIGKDD Explorations Newsletter 24.2 (2022): 61-77.
> >
> > [5] Zhou, Jiajun, Jie Shen, and Qi Xuan. "Data augmentation for graph classification." Proceedings of the 29th ACM International Conference on Information & Knowledge Management. 2020.
> >
> > [6] Zhu, Yanqiao, et al. "Graph contrastive learning with adaptive augmentation." Proceedings of the Web Conference 2021. 2021.
> >
> > [7]  Liu, Yixin, et al. "Graph self-supervised learning: A survey." IEEE Transactions on Knowledge and Data Engineering 35.6 (2022): 5879-5900.
> >
> > [8] Zhao, Tong, et al. "Graph data augmentation for graph machine learning: A survey." arXiv preprint arXiv:2202.08871 (2022).

---

> > > ### Author Response · Authors · 2023-08-17
> > > **We've Carefully Addressed Each of the Questions You Raised and Eagerly Hoping for Your Valuable Feedback**
> > >
> > > Dear Reviewer 8yVE,
> > >
> > > Thank you deeply for your thoughtful review and valuable insights. We've taken every question you've raised to heart and have responded in detail where needed. We sincerely hope you'll take a moment to reflect on our responses, trusting that they meet your considerations. Your time and expertise in reviewing our work mean so much to us.
> > >
> > > Warm Regards,
> > >
> > > Authors

---

> > > > ### Comment · Area_Chair_TEXL · 2023-08-18
> > > > **Pls provide your comments to authors' rely  - 8yVE**
> > > >
> > > > The authors have diligently crafted a rebuttal, offering comprehensive insights to address your inquiries. Your timely response would be greatly valued to ensure effective progress and resolution. Note that discussion with authors ends next Monday (August 21st).
> > > >
> > > > Can 8yVE provide your response soon?
> > > >
> > > > Thanks,
> > > > AC

---

> > > > > ### Comment · Area_Chair_TEXL · 2023-08-18
> > > > >
> > > > > Dear Reviewers,
> > > > >
> > > > > I hope this message finds you well. First and foremost, thank you all for your valuable contributions to the NeurIPS review process.
> > > > >
> > > > > As we approach a critical phase of the review cycle, we've noticed that there are still pending responses to authors' rebuttals and specific replies from some reviewers. These responses are pivotal in ensuring a comprehensive and constructive review process for all submissions.
> > > > >
> > > > > In light of this, we kindly request those of you who have outstanding responses to kindly prioritize completing them as soon as possible. Your prompt attention to this matter would be greatly appreciated, as it will help us maintain the momentum of the review process and ensure that authors receive timely feedback.
> > > > >
> > > > > If you encounter any challenges or require additional assistance, please don't hesitate to reach out to us. We are here to support you and facilitate a smooth and efficient review process.
> > > > >
> > > > > Once again, we express our gratitude for your dedication to NeurIPS and your role as a reviewer. Your efforts play a vital role in shaping the quality and impact of the conference.
> > > > >
> > > > > Thank you for your attention, and we look forward to your continued collaboration.
> > > > >
> > > > > Thanks,
> > > > > AC

---

> > > > > > ### Comment · Reviewer_8yVE · 2023-08-20
> > > > > >
> > > > > > I thank the authors for their efforts in answering my questions, and my concerns are mostly addressed. Considering the strong empirical results and the novelty of the method, I increase my rating to weak accept. However, as I'm very unfamiliar with the field of node classification and the corresponding benchmark, I keep my original confidence rating.

---

### Official Review · Reviewer_zTQi · 2023-07-01

**Soundness:** 3 good
**Presentation:** 2 fair
**Contribution:** 3 good
**Rating:** 6
**Confidence:** 3

**Summary:**

This paper is majoring on the imbalance problem in graph node classification. The authors confirmed the relationship between model variance and the degree of dataset imbalance by adopting the Bias-Variance Decomposition. Furthermore, they diverted a regularization term for approximating the variance of the model from the above theoretical analysis.

**Strengths:**

This paper discovers the relation between imbalance and variance, estimates the variance with graph argumentation, and solves the imbalance problem by adopting two proposed regularization terms, which is an important problem in node classification.
The regularization term is theoretically derived from the Bias-variance Decomposition, and the author, for the first time, fits this theory into the field of imbalance by making two weak assumptions, which is a good contribution to this field.

**Weaknesses:**

- Overall, this paper contains many typos and abuse of the notation that may make the reader uncomfortable. I list those points in the limitations.
- In this work, the authors make a strong assumption that all embeddings follow the multivariate normal distribution with the same sample variance ($h(x)^T\Lambda^ih(x)$ in the paper) between each class. And the authors use this assumption to derive the core relation between variance and imbalance, then the regularization term.
However, this assumption is strongly related to the quality of the extracted embeddings.
One important guarantee for this assumption is a well-trained feature extractor with a clear boundary between each class.
However, at the initial stage, the model is not well-trained, meaning that a large variance of $h(x)^T\Lambda^ih(x)$ in each class.
It is unclear how this will affect the power of regularization though the overall results look good, and the authors did not make a discussion of the above problem in the paper.

**Questions:**

1. Why do the authors assume node embeddings follow the multivariate normal distribution?
2. Could the authors provide the significance of the results in Fig1 (e.g., using hypothesis testing)?


**Limitations:**

The author could improve their manuscript by solving the below points and typos:
- line 25-27: sentence should be shortened.
- line 76: no definition for $N$, $F$ and $D$.
- Section 3.1: Is there a lack of citations of Bias-variance Decomposition?
- The term between Eq(2) and line 111, line 114 is different.
- line 140: "condition on" should be replaced by "conditioning on".
- line 208: indicator function is often denoted as $\\mathbf{1}_{ \\{ \\cdots \\} }$.

---

> ### Author Rebuttal · Authors · 2023-08-08
>
> We sincerely thank you for the comments. We appreciate your comments that the theoretical and experimental contribution of RVGNN is strong for tackling imbalance problems. We address all your concerns below:
> ***
> **Q1**: This paper contains many typos and abuse of the notation that may make the reader uncomfortable.
>
> **Answer1**:
>
> Thanks for highlighting the concerns related to typos and notation inconsistencies. We have **taken every point you listed into consideration. We have thoroughly revised the paper, making sure to correct all the typos and standardize the notation throughout**. Also based on this, we have **updated the notation table (Appendix G)** of the full text we compiled earlier to facilitate the reading of the article. We believe these changes have enhanced the readability and clarity of the manuscript. **Details are as follows:**
>
> - For the line 25-27 which sentence should be shortened, we modified it to: "Graph-structured data often requires consideration of the data's topology and environment, and empirical evidence indicates that topological asymmetries can also affect the performance of models."
> ***
> - For the line 76 which is no definition for N, F, D, we modified it to: "N, F, and D are the dimensions of features and models, and they have the same meaning as n, f, and d of Preliminaries. We unify them."
> ***
> - For Section 3.1 which is a lack of citations of Bias-variance Decomposition, we have cited some of the most relevant papers[1,2,3,4].
>
>   [1] Belkin, Mikhail, et al. "Reconciling modern machine-learning practice and the classical bias–variance trade-off." Proceedings of the National Academy of Sciences 116.32 (2019): 15849-15854
>
>   [2] Kohavi, Ron, and David H. Wolpert. "Bias plus variance decomposition for zero-one loss functions." ICML. Vol. 96. 1996.
>
>   [3] Von Luxburg, Ulrike, and Bernhard Schölkopf. "Statistical learning theory: Models, concepts, and results." Handbook of the History of Logic. Vol. 10. North-Holland, 2011. 651-706.
>
>   [4] Neal, Brady. "On the bias-variance tradeoff: Textbooks need an update." arXiv preprint arXiv:1912.08286 (2019).
> ***
> - For the term between Eq(2) and line 111, line 114 is different, I guess you are confused about the definition of $x_{i}$ in line 111, for a better understanding for the readers, we modified the sentence as " We make two assumptions in our approach. Firstly, we assume that the node embeddings $h^i$  of node $x^i$ extracted by a graph neural network for nodes belonging to class $i$ follow a multivariate normal distribution $h^{i} \sim N(\mu^{i}, \Lambda^{i})$ ''. For $h_{n_{i}}^{i}$, it means the embedings of the $n_{i}$th node belong to $i$ class. We know that this notation can also cause some confusion, and we have carefully corrected it in the paper.
> ***
> - For the line 140 where "condition on" should be replaced by "conditioning on", we have corrected it.
> ***
> - For the line 208 that indicator function is often denoted as $\boldsymbol{1}_{\{...\}}$, we have followed your idea and have corrected it.
> ***
> Please let us know if there are any other aspects that need further attention.
> ***
> **Q2**: The strong assumption that all embeddings follow the multivariate normal distribution with the same sample variance $(h(x)^{T}\Lambda^{i}h(x))$ in the paper) between each class. And the authors use this assumption to derive the core relation between variance and imbalance, then the regularization term.
>
> **Answer2**:
>
> We appreciate the reviewer's feedback. Our work indeed assumes, for illustrative purposes, that different classes share the same $\Lambda^i$ when demonstrating the relationship between variance and imbalance. However, it is vital to clarify that our regularization term is not contingent on this assumption. By employing graph augmentation, we generate pseudo-node pairs belonging to an identical class, subsequently reducing the feature discrepancy within these pairs to diminish the class-specific variance. It's worth noting that this variance approximation remains unaffected by the discrepancy of different $\Lambda^i$ values. The optimization is conducted concurrently with the classification training. As you astutely pointed out, by the conclusion of the training phase, the relationship between variance and dataset imbalance is evident. Given that our algorithm consistently yields a model with reduced variance, we confidently assert that the resultant model exhibits robustness in imbalanced scenarios.
> ***
> **Q3**: Could the authors provide the significance of the results in Fig1 (e.g., using hypothesis testing)?
>
> **Answer3**:
>
> Thank you for your constructive suggestions. **We computed the Pearson correlation coefficient between variance and imbalance ratio (log), as presented in the table below**. The Pearson correlation coefficient, denoted as $r$, is a prevalent metric for gauging linear correlations. This coefficient lies between $-1$ and $1$ and reflects both the magnitude and direction of the correlation between two variables. An $r$ value greater than $0.5$ indicates a strong positive correlation. Furthermore, the p-value results from a hypothesis test with the null hypothesis $H_{0}: \rho = 0$ and the alternative hypothesis $H_{a}: \rho \neq 0$, where $\rho$ represents the population correlation coefficient.
>
> |            | Citeseer-GCN | Citeseer-GAT | Citeseer-SAGE | PubMed-GCN | PubMed-GAT | PubMed-SAGE |
> |------------|--------------|--------------|---------------|------------|------------|-------------|
> | **$r$**    | 0.751        | 0.786        | 0.642         | 0.694      | 0.760      | 0.747       |
> | **P-value**| 3.203e-14    | 1.516e-14    | 5.107e-07     | 2.233e-09  | 2.344e-14  | 5.634e-13   |
>
> **Given that the Pearson correlation coefficient between variance and imbalance ratio exceeds $0.5$, and the p-value is below $0.01$, we deduce that there is a robust correlation between variance and imbalance ratio. This relationship is statistically significant at the $0.01$ significance level.**

---

> > ### Comment · Reviewer_zTQi · 2023-08-10
> > **I'll raise my score.**
> >
> > The authors clearly addressed my concern. Thus, I will raise my score as 6.

---

> > > ### Author Response · Authors · 2023-08-11
> > > **Expressing Gratitude and Seeking Further Consideration**
> > >
> > > Thank you for your constructive feedback and the subsequent score adjustment. **We truly value your insights and are pleased that our rebuttal addressed your concerns.**
> > >
> > > We earnestly believe that **our research provides a notable contribution to the domain, which is the first to concentrate on optimizing the model's variance, both in graph imbalance learning and traditional imbalanced learning.** Given your expertise, we would be grateful if you could **reconsider areas of our work that might further align with the conference's criteria**, possibly leading to an even more favorable assessment.
> > >
> > > Once again, we deeply appreciate your time and thoughtful evaluation. We eagerly await your final review.
> > >
> > > Best regards

---

### Official Review · Reviewer_j7Kz · 2023-07-05

**Soundness:** 3 good
**Presentation:** 3 good
**Contribution:** 3 good
**Rating:** 4
**Confidence:** 4

**Summary:**

This paper focuses on semi-supervised imbalanced node
classification tasks. Specifically, the authors first establish a theoretical result that connects the imbalance ratio with the model variance and then propose a new regularization term related to the variance based on the graph augmentation technique. Experimental results show the effectiveness of the proposal.

**Strengths:**

1. The authors give theoretical results that connect the imbalance ratio to the model variance. The results are insightful.
2. The experimental results show the proposal can achieve the SOTA performance on various class-imbalanced datasets.

**Weaknesses:**

1. The theoretical results rely on strong assumptions, which may be difficult to satisfy in real-world tasks.
2.  From Figure 3, it looks like the performance is highly influenced by the hyper-parameters. So how to determine the hyper-parameter should be discussed in the paper.
3. Moreover, I doubt that the performance of the proposal is not related to the theoretical results, it may rely on graph augmentation. Thus, more ablation studies should be conducted.

**Questions:**

1. How to determine the hyper-parameters in different datasets?
2. How to demonstrate the superiority of algorithm performance is related to theoretical results?

**Limitations:**

Yes.

---

> ### Author Rebuttal · Authors · 2023-08-08
>
> We sincerely thank you for the comments. We appreciate your comments that the theoretical and experimental superiority is strong for our work. We address all your concerns below:
> ***
> **Q1**: The theoretical results rely on strong assumptions, which may be difficult to satisfy in real-world tasks.
>
> **Answer1**:
>
> We acknowledge and appreciate your observation regarding our assumption about the embeddings following a multivariate normal distribution to approximate the variance. **Our rationale is inspired by the Central Limit Theorem (CLT)**. The generation of these embeddings of GNN involves aggregating information from several random processes—such as random walks and aggregations from neighboring nodes[1,2]. Each of these can be viewed as a distinct random variable. As we aggregate numerous such variables, the CLT suggests that their aggregate effect tends to approach a normal distribution, given they possess finite expectations and variances.
>
> **We recognize this as an approximation and aimed to streamline our theoretical framework**. In practice, this assumption has proven to be a satisfactory fit. Nonetheless, we are cognizant of its limitations and value your feedback on this matter.
>
>
> [1] Cavallari, Sandro, et al. "Learning community embedding with community detection and node embedding on graphs." Proceedings of the 2017 ACM on Conference on Information and Knowledge Management. 2017.
>
> [2] Xu, Mengjia. "Understanding graph embedding methods and their applications." SIAM Review 63.4 (2021): 825-853.
>
> ***
> **Q2**: From Figure 3, it looks like the performance is highly influenced by the hyper-parameters. So how to determine the hyper-parameter should be discussed in the paper.
>
> **Answer2**:
>
> - We would like to **kindly draw your attention to Appendix E.4 of the original submission**, where we have provided detailed range for our hyperparameter search.
>
> - In determining the hyperparameters for different datasets, we follow a rigorous and systematic approach rather than relying on heuristic methods. **Our methodology involves conducting a hyperparameter sweep in the conventional manner**.
>
>   To begin, we partition the dataset into training, validation, and test sets. During the hyperparameter sweep, we carefully configure the range for each hyperparameter and employ some strategies to generate sets of hyperparameters. **We then search for the set of hyperparameters that yields the best performance on the validation set, using F1 score as key evaluation metrics**. In our experiments, we observed that **the hyperparameter selection process for our model is exceptionally robust.**
>
>   To facilitate this process, we leverage the wandb platform (wandb.ai) to organize our experiments and **utilize the built-in Bayes strategy within the hyperparameter sweep**. This allows for efficient exploration of the hyperparameter space, enabling us to identify the optimal combination of hyperparameters for each dataset.
>
> ***
> **Q3**: Moreover, I doubt that the performance of the proposal is not related to the theoretical results, it may rely on graph augmentation. Thus, more ablation studies should be conducted.
>
> **Answer3**:
> - The core idea of our work is to optimize the variance of the model. As the key technology we use to **approximate the variance, graph data augmentation plays a very important role**. In fact, the **theoretical result derivation also does depend on graph data augmentation**.
>
> - We are not sure that we understand your point. We think that your concern may be about that the use of contrastive loss ($L_{IR}$) together with data augmentation plays a key role. Through **the following detailed ablation analysis**, we clearly show that $L_{VR}$  plays the most important role.
>
> |                  | **Dataset**          | **CiteSeer-Semi**    | **PubMed-Semi**      | **Computers-Semi**    |
> |:-----------------|:--------------------:|:--------------------:|:--------------------:|:--------------------:|
> |                  |                      | F1-Score(%)          | F1-Score(%)          | F1-Score(%)           |
> |------------------|----------------------|----------------------|----------------------|----------------------|
> | **SAGE**         | Sup                  | 44.43                | 64.80                | 77.16                 |
> |                  | Sup+IR               | 60.92                | 68.35                | 78.62                 |
> |                  | Sup+VR               | 62.33                | 73.68                | 79.54                 |
> |                  | Sup+IR+VR            | 64.91                | 76.44                | 80.40                 |
> |------------------|----------------------|----------------------|----------------------|----------------------|
> | **GAT**          | Sup                  | 48.08                | 65.91                | 74.04                 |
> |                  | Sup+IR               | 65.35                | 67.04                | 74.35                 |
> |                  | Sup+VR               | 65.67                | 72.87                | 77.65                 |
> |                  | Sup+IR+VR            | 65.70                | 74.46                | 78.12                 |
> |------------------|----------------------|----------------------|----------------------|----------------------|
> | **GCN**          | Sup                  | 43.98                | 62.28                | 73.54                 |
> |                  | Sup+IR               | 53.94                | 66.16                | 76.32                 |
> |                  | Sup+VR               | 57.43                | 72.34                | 77.09                 |
> |                  | Sup+IR+VR            | 59.73                | 74.48                | 78.49                 |
> |------------------|----------------------|----------------------|----------------------|----------------------|

---

> > ### Author Response · Authors · 2023-08-18
> > **We've Carefully Addressed Each of the Questions You Raised and Eagerly Hoping for Your Valuable Feedback**
> >
> > Dear Reviewer j7Kz,
> >
> > From the depths of our hearts, we express our sincerest gratitude for your thoughtful and perceptive review. Every question you posed has been carefully contemplated, and we've strived to offer thorough answers where appropriate. We cordially invite you to peruse our responses, with the hope that they align with your insights. The time and wisdom you've invested in reviewing our work profoundly moves us.
> >
> > Warm Regards,
> >
> > Authors

---

### Official Review · Reviewer_vtyV · 2023-07-06

**Soundness:** 2 fair
**Presentation:** 3 good
**Contribution:** 2 fair
**Rating:** 5
**Confidence:** 3

**Summary:**

This paper introduces a new approach to address the issue of class imbalance in graph neural networks (GNNs) for learning on graph-structured data. It also provides a novel theoretical perspective for addressing the problem of imbalanced node classification in GNNs.

**Strengths:**

1 The article is well written and easy to understand.

3 The authors prove their claim by theoretical derivation.

4 The experimental results further corroborate the authors' view.


**Weaknesses:**

1 The use of L to represent the set of labeled nodes is still relatively rare in the definition of graphs, and the authors are advised to describe the definition involved further in the opening paragraph of the method.

2 Is the authors considering restructuring the article? Putting the theoretical derivation before the introduction of the method may not give the reader a good reading experience.

3 I am curious about this one: the authors point out that the full graph noise distribution is taken into account when sampling the graph in order to sample the training set that has approximate variance. However, if one considers graph learning in an open world, then the noise from the added data might be quite different from the original graph data.

4 In Table 1, the authors can consider further reporting the percentage of performance improvement compared to SOTA.

5 In Fig 3(a), it seems that the performance improvement from VR loss is not significant. Are you considering the Sup+VR combination? Also, can you further diet why VR loss gives a smaller boost to GAT?


**Questions:**

No more questions.

---

> ### Author Rebuttal · Authors · 2023-08-08
>
> We sincerely thank you for the comments. We appreciate your comments that the theoretical and experimental superiority is strong for our model. We address all your concerns below:
> ***
> **Q1**: The use of L to represent the set of labeled nodes is still relatively rare in the definition of graphs, and the authors are advised to describe the definition involved further in the opening paragraph of the method.
>
> **Answer1**:
>
> We have modified our notation to **use $V_{L}$ and $V_{U}$**  to represent the sets of labeled and unlabeled nodes, respectively, replacing the previous $L$ and $U$. This revised notation aligns better with common conventions.
> ***
> **Q2**: Is the authors considering restructuring the article? Putting the theoretical derivation before the introduction of the method may not give the reader a good reading experience.
>
> **Answer2**:
>
> We sincerely appreciate your valuable feedback concerning the structure of our paper. We will **thoughtfully consider your suggestions and discuss potential refinements to the article's organization** with our co-authors. Our goal is to enhance the clarity and readability for the audience.
> ***
> **Q3**: I am curious about this one: the authors point out that the full graph noise distribution is taken into account when sampling the graph in order to sample the training set that has an approximate variance. However, if one considers graph learning in an open world, then the noise from the added data might be quite different from the original graph data.
>
> **Answer3**:
>
> Thank you for raising this intriguing scenario. Our original focus **revolved around semi-supervised learning within the "close-world" setting, where all labeled and unlabeled nodes are accessible during the training stage**. Therefore, we would like to clarify that the concern you've mentioned won't be applicable to our current paper.
> We do recognize that in the "open-world" setting, the regularization term introduced in our paper **may not be able to handle the variance of unseen data effectively**. However, it still serves as a valuable regularization term for addressing the imbalance present within the known dataset. We appreciate your acknowledgment of the potential value in estimating variance within the "open-world" setting, and we agree that it could pave the way for generalizing our algorithm.
> ***
> **Q4**: In Table 1, the authors can consider further reporting the percentage of performance improvement compared to SOTA.
>
> **Answer4**:
>
> We agree that including the percentage of performance improvement compared to the state-of-the-art (SOTA) would provide additional clarity. **We have updated the tables** to reflect this comparison and believe it enhances the understanding of our method's effectiveness.
>
> ***
> **Q5**: In Fig 3(a), it seems that the performance improvement from VR loss is not significant. Are you considering the Sup+VR combination? Also, can you further diet why VR loss gives a smaller boost to GAT?
>
> **Answer5**:
>
> 1. Thank you for pointing that out. It's true that the notable performance enhancement from the VR loss isn't as pronounced in the CiteSeer-GAT instance. **However, in a majority of the other cases, the role of $L_{VR}$ is quite significant**.  We would like to **kindly draw your attention to Appendix D of the original submission**, where we have provided more detailed ablation experiments across additional datasets like PubMed and Computers, and using various model scenarios such as GCN, GAT, and SAGE. We hope this addresses your concerns.
>
> 2. Based on your insightful feedback, we **rigorously incorporated the combination of Sup+$L_{VR}$ into our ablation study as follows**. The results affirm a fundamental insight of our work: optimizing the model's variance proves to be an effective method to address the long-tail problem.
>
> |                  | **Dataset**          | **CiteSeer-Semi**    | **PubMed-Semi**      | **Computers-Semi**    |
> |:-----------------|:--------------------:|:--------------------:|:--------------------:|:--------------------:|
> |                  |                      | F1-Score(%)          | F1-Score(%)          | F1-Score(%)           |
> |------------------|----------------------|----------------------|----------------------|----------------------|
> | **SAGE**         | Sup                  | 44.43                | 64.80                | 77.16                 |
> |                  | Sup+IR               | 60.92                | 68.35                | 78.62                 |
> |                  | Sup+VR               | 62.33                | 73.68                | 79.54                 |
> |                  | Sup+IR+VR            | 64.91                | 76.44                | 80.40                 |
> |------------------|----------------------|----------------------|----------------------|----------------------|
> | **GAT**          | Sup                  | 48.08                | 65.91                | 74.04                 |
> |                  | Sup+IR               | 65.35                | 67.04                | 74.35                 |
> |                  | Sup+VR               | 65.67                | 72.87                | 77.65                 |
> |                  | Sup+IR+VR            | 65.70                | 74.46                | 78.12                 |
> |------------------|----------------------|----------------------|----------------------|----------------------|
> | **GCN**          | Sup                  | 43.98                | 62.28                | 73.54                 |
> |                  | Sup+IR               | 53.94                | 66.16                | 76.32                 |
> |                  | Sup+VR               | 57.43                | 72.34                | 77.09                 |
> |                  | Sup+IR+VR            | 59.73                | 74.48                | 78.49                 |
> |------------------|----------------------|----------------------|----------------------|----------------------|

---

> > ### Author Response · Authors · 2023-08-18
> > **We've Carefully Addressed Each of the Questions You Raised and Eagerly Hoping for Your Valuable Feedback**
> >
> > Dear Reviewer vtyV,
> >
> > From the bottom of our hearts, we thank you for your kind and insightful review. Each question you brought up has been tenderly considered, and we've endeavored to provide comprehensive answers where necessary. We warmly invite you to take a moment to go through our responses, hoping they resonate with your thoughts. The time and wisdom you've shared in reviewing our work touches us deeply.
> >
> > Warm Regards,
> >
> > Authors

---

> > ### Comment · Reviewer_vtyV · 2023-08-19
> >
> > Thank you for the response. Based on the authors' rebuttal and comments from other reviewers, I decided to keep my score.

---

### Official Review · Reviewer_bhDb · 2023-07-09

**Soundness:** 2 fair
**Presentation:** 2 fair
**Contribution:** 1 poor
**Rating:** 5
**Confidence:** 3

**Summary:**

This paper proposes a theory that relates data imbalance to model variance and designs a method to mitigate the bias of class imbalance.
For the theory, this paper finds that the variance of each class is proportional to the invert of the number of samples in that class.
For the method, this paper uses a regularization term to approximate the model variance and construct a varied training distribution with graph augmentation. The regularization term is added to the original loss function.
 Results on public-split datasets and naturally imbalanced datasets verify the proposal.

**Strengths:**

This paper first connects the model variance and class imbalance in graph learning.

**Weaknesses:**

1. L_IR objective is more closely related to variance than contrastive learning. It would help to clarify the difference between your work and contrastive graph learning.

2. This paper does not discuss previous work on variance and imbalance in traditional imbalanced learning. Since the variance calculated in this paper does not use graph-specific settings, it is not unique to graph learning.

**Questions:**

1. Why can you assume that node embeddings follow a multivariate normal distribution?

2. Many elements of Figure 2 need descriptions. E.g.,  What do yellow and blue circles mean?

**Limitations:**

This paper does not discuss limitations. But I think the novelty and unique contribution is limited.

---

> ### Author Rebuttal · Authors · 2023-08-08
>
> We sincerely thank you for the comments. We appreciate your comments that our novelty that first connects the model variance and class imbalance problem on the graph. We address all your concerns below:
> ***
> **Q1**: $L_{IR}$  objective is more closely related to variance than contrastive learning. It would help to clarify the difference between your work and contrastive graph learning.
>
> **Answer1**:
> - The main focus of our study is to utilize graph augmentation to formulate the loss function $L_{VR}$ for approximating the model's variance. It's worth clarifying that **$L_{IR}$ is not directly linked to the model's variance**. We hope this provides a clearer understanding.
> - We acknowledge that there might have been confusion caused by a misleading sentence in Sec 4.2, where we mentioned, "In this section, we propose an extension to the concept of graph contrastive learning that emphasizes the invariance of node representations in semi-supervised scenarios." We understand that this statement could imply a direct relation between $L_{IR}$ and variance, which was not our intention. **In the revision, we will carefully rephrase this section to clarify our intentions** and avoid any potential misinterpretations.
> - For the term $L_{IR}$, our experiments revealed that the inclusion of an additional loss term, $L_{IR}$, **is beneficial in ensuring that the graph augmentation does not negatively influence GNN feature extraction**. This concept is inspired by graph contrastive learning, which we have tailored to better fit our specific setting.
>
> ***
> **Q2**: This paper does not discuss previous work on variance and imbalance in traditional imbalanced learning. Since the variance calculated in this paper does not use graph-specific settings, it is not unique to graph learning.
>
> **Answer2**:
> - First, we truly appreciate the reviewer's attention and feedback about this concern. After an in-depth literature review, we found that our study is among **the first to concentrate on optimizing the model's variance, both in graph learning and traditional imbalanced learning**. We consider this a significant contribution and have approached the challenge from a thoughtful theoretical perspective. As mentioned in Section 6, our future directions include extending RVGNN and its foundational theories to the realms of computer vision and natural language processing. Considering this focus, we feel that an extensive discussion about the combination of variance and imbalance in traditional imbalanced learning might not be central to our current study. Once again, thank you for your invaluable insights and suggestions.
> - Regarding your second concern, we would like to clarify that in our work, we indeed incorporate graph-specific settings when approximating the variance of the model.
>   - In optimizing the approximate variance, graph data augmentation(GDA) is crucial as it serves to emulate diverse training sets. In this work, it's worth noting that GDA is distinct from traditional data augmentation. Here, **GDA acts as a form of dataset augmentation**.  It not only perturbs the features of individual nodes but also significantly modifies the graph dataset by cropping or adding edges. However, we recognize that **applying our framework to traditional settings may not be direct**.
>   - As detailed in our paper, we believe that **addressing the graph imbalance problem by optimizing the model's variance offers a more meaningful approach compared to traditional imbalanced learning domains**. Unlike conventional imbalanced learning, devising GNN imbalanced learning methods for graph-structured data presents distinct challenges. Graph-structured data often necessitate careful consideration of the data’s topology and surroundings. Traditional strategies such as oversampling and loss function engineering often fall short in achieving satisfactory results. More specifically, the modeling approach for data personalization shows marked deficiencies in scalability and generalization capability. Therefore, a more fundamental and theoretical viewpoint is urgently needed to address the imbalance node classification issue. In this work, we introduce a novel perspective to comprehend graph imbalance through the prism of Bias-Variance Decomposition. This forms the motivation for our work.
>
> ***
> **Q3**: The node embeddings follow a multivariate normal distribution.
>
> **Answer3**:
>
> We acknowledge and appreciate your observation regarding our assumption about the embeddings following a multivariate normal distribution. **Our rationale is inspired by the Central Limit Theorem (CLT)**. The generation of these embeddings of graph involves aggregating information from several random processes—such as random walks and aggregations from neighboring nodes[1,2]. Each of these can be viewed as a distinct random variable. As we aggregate numerous such variables, the CLT suggests that their aggregate effect tends to approach a normal distribution, given they possess finite expectations and variances.
>
> **We recognize this as an approximation and aimed to streamline our theoretical framework**. In practice, this assumption has proven to be a satisfactory fit. Nonetheless, we are cognizant of its limitations and value your feedback on this matter.
>
> [1] Cavallari, Sandro, et al. "Learning community embedding with community detection and node embedding on graphs." Proceedings of the 2017 ACM on Conference on Information and Knowledge Management. 2017.
>
> [2] Xu, Mengjia. "Understanding graph embedding methods and their applications." SIAM Review 63.4 (2021): 825-853.
> ***
> **Q4**: Many elements of Figure 2 need descriptions. E.g., What do yellow and blue circles mean??
>
> **Answer4**:
>
> We have revised Figure 2 to provide more detail. In this figure, additional elements have been incorporated that were not previously marked  (yellow and blue represent different classes).  **We have placed the modified Figure 2 to in the PDF file that has been uploaded.**

---

> > ### Author Response · Authors · 2023-08-18
> > **We've Carefully Addressed Each of the Questions You Raised and Eagerly Hoping for Your Valuable Feedback**
> >
> > Dear Reviewer bhDb,
> >
> > Thank you profoundly for your insightful review and invaluable feedback. We have earnestly considered each query you posed and have addressed them in depth where necessary. We genuinely hope that you will take some time to review our detailed responses, confident that they address your concerns. Your expertise and the time you've invested in reviewing our work are deeply appreciated by us.
> >
> > Warm Regards,
> >
> > Authors

---

> > > ### Comment · Reviewer_bhDb · 2023-08-21
> > > **Thank you for your rebuttal.**
> > >
> > > I have read your response to me and other reviewers. In the rebuttal, the authors clarify the story and the technique issues. Since the paper still has limited contribution. I decided to raise the score from 4 to 5, hoping the authors could include them.

---

### Author Rebuttal · Authors · 2023-08-09

We thank all the reviewers for their constructive and insightful feedback.
  - We have thoroughly revised the paper, making sure to correct all the typos and standardize the notation throughout.
  - We **have uploaded our revised Figure 2(Pipeline of RVGNN) here**, in this figure, additional elements have been incorporated that were not previously marked.

---

### Author Response · Authors · 2023-08-17
**Further Clarification (1/2) about the Underlying Assumptions of this Work for Reviewer bhDb, j7Kz**

## Assumptions that node embeddings follow a multivariate normal distribution
-  **Standing on the Shoulders of Giants:**  The adoption of the multivariate normal distribution assumption is primarily anchored in the Central Limit Theorem (CLT). We can notice there already have influential works [1, 2, 3, 4, 5, 6] in the GNN domain that have similarly leveraged this assumption for their theoretical analyses and derivations, a few of which we have enumerated in our manuscript (as referenced). We have duly cited these pioneering works in the appropriate sections in our revised version, thus ensuring that our adoption of this assumption does not appear arbitrary or unwarranted.

     [1] Hajiramezanali, Ehsan, et al. "Variational graph recurrent neural networks." Advances in neural information processing systems 32 (2019).

     [2] Shi, Min, et al. "Multi-class imbalanced graph convolutional network learning." Proceedings of the Twenty-Ninth International Joint Conference on Artificial Intelligence (IJCAI-20). 2020.

    [3] Chen, Kaixuan, et al. "Distribution knowledge embedding for graph pooling." IEEE Transactions on Knowledge and Data Engineering (2022).

   [4] Funke, Thorben, et al. "Low-dimensional statistical manifold embedding of directed graphs." arXiv preprint arXiv:1905.10227 (2019).''

   [5] Cavallari, Sandro, et al. "Learning community embedding with community detection and node embedding on graphs." Proceedings of the 2017 ACM on Conference on Information and Knowledge Management. 2017.

   [6] Xu, Mengjia. "Understanding graph embedding methods and their applications." SIAM Review 63.4 (2021): 825-853.

***
- **Striking a Balance:** We acknowledge that this assumption might not mirror the intricate realities perfectly. However, it offers a means to streamline our theoretical framework. Building on this foundation, we introduced Lemma 1, which provides our $L_{VR}$  loss term, acting as the model's variance estimation component.

***
- **Empirical Validation:**  In light of the discussions, we've undertaken a more comprehensive ablation study during the rebuttal phase. This study sheds light on the efficacy of the model variance estimation term derived from this assumption. We believe our findings will underscore its merit and the practical relevance of the choices we've made.

|                         | **Dataset**          | **CiteSeer-Semi**      | **PubMed-Semi**       | **Computers-Semi**    |
|:-----------------------:|:--------------------:|:----------------------:|:----------------------:|:----------------------:|
| **SAGE**                |                      | F1-Score(%)            | F1-Score(%)            | F1-Score(%)            |
|                         | Sup                  | 44.43                  | 64.80                  | 77.16                  |
|                         | Sup+IR               | 60.92                  | 68.35                  | 78.62                  |
|                         | Sup+VR               | 62.33                  | 73.68                  | 79.54                  |
|                         | Sup+IR+VR            | 64.91                  | 76.44                  | 80.40                  |
| **GAT**                 |                      |                        |                        |                        |
|                         | Sup                  | 48.08                  | 65.91                  | 74.04                  |
|                         | Sup+IR               | 65.35                  | 67.04                  | 74.35                  |
|                         | Sup+VR               | 65.67                  | 72.87                  | 77.65                  |
|                         | Sup+IR+VR            | 65.70                  | 74.46                  | 78.12                  |
| **GCN**                 |                      |                        |                        |                        |
|                         | Sup                  | 43.98                  | 62.28                  | 73.54                  |
|                         | Sup+IR               | 53.94                  | 66.16                  | 76.32                  |
|                         | Sup+VR               | 57.43                  | 72.34                  | 77.09                  |
|                         | Sup+IR+VR            | 59.73                  | 74.48                  | 78.49                  |

---

> ### Author Response · Authors · 2023-08-17
> **Further Clarification (2/2) about the Underlying Assumptions of this Work for Reviewer bhDb, j7Kz**
>
> - **Clarifying Assumptions about Sample Variance**: We discerned concerns regarding our assumption that each class exhibits uniform sample variance. This particular assumption was instrumental in deriving our Theorem 1, which posits a direct correlation between model variance and the degree of data imbalance. However, it is pivotal for us to articulate that this assumption, while vital for validating Theorem 1, is not imperative for the derivation of the $L_{VR}$ loss term. To substantiate the aforementioned positive correlation, we employed a rigorous hypothesis testing procedure.
>
> We computed the Pearson correlation coefficient between variance and imbalance ratio (log), as presented in the table below. The Pearson correlation coefficient, denoted as $r$, is a prevalent metric for gauging linear correlations. This coefficient lies between \(-1\) and $r$ and reflects both the magnitude and direction of the correlation between two variables. An \(r\) value greater than \(0.5\) indicates a strong positive correlation. Furthermore, the p-value results from a hypothesis test with the null hypothesis $(H_{0}: \rho = 0)$ and the alternative hypothesis $(H_{a} : \rho \neq 0)$, where $(\rho)$ represents the population correlation coefficient.
>
>
>
> |            | Citeseer-GCN | Citeseer-GAT | Citeseer-SAGE | PubMed-GCN | PubMed-GAT | PubMed-SAGE |
> |------------|--------------|--------------|---------------|------------|------------|-------------|
> | **$r$**    | 0.751        | 0.786        | 0.642         | 0.694      | 0.760      | 0.747       |
> | **P-value**| 3.203e-14    | 1.516e-14    | 5.107e-07     | 2.233e-09  | 2.344e-14  | 5.634e-13   |
>
> Given that the Pearson correlation coefficient between variance and imbalance ratio exceeds \(0.5\), and the p-value is below \(0.01\), we deduce that there is a robust correlation between variance and imbalance ratio. This relationship is statistically significant at the \(0.01\) significance level.
> ***
> Additionally, we would like to highlight that Reviewer zTQi（https://openreview.net/forum?id=0gvtoxhvMY&noteId=V0kRoHvvAu, https://openreview.net/forum?id=0gvtoxhvMY&noteId=4vRD8nGnhm ) has expressed agreement with our justification for this particular assumption. While we deeply value and respect the diverse opinions of all reviewers, we believe that this concurrence further underscores the validity and the thought process behind our approach.
>
> We understand that the realm of research thrives on diverse perspectives, and it is through rigorous scrutiny and collective feedback that we refine our work. Our intention is not to use this as a definitive validation, but rather as an indication that our reasoning resonates with some expert opinions in the field. We remain committed to addressing all concerns and clarifications, ensuring our research is both comprehensive and well-understood.
>
> ***
> It's pivotal for us to stress that these assumptions serve as foundational pillars for theoretical development, and while they may seem restrictive, they have been chosen after judicious consideration. They play a role in allowing us to distill complex relationships and provide more tractable insights, all while being grounded in empirical efficacy. Your rigorous assessment and poignant queries are indeed instrumental in helping us elucidate our methodological choices better. We remain thankful for the thoughtful feedback, which pushes our research boundaries and ensures a comprehensive exposition.

---

### Decision · Program_Chairs · 2023-09-21

**Decision:**

Accept (poster)

**Comment:**

The paper has received positive and constructive feedback from multiple reviewers. The innovative approach of connecting model variance with class imbalance in graph learning has been recognized for its novelty and significance. The comprehensive rebuttal process effectively addressed concerns across various aspects of the research, such as theoretical derivations, experimental results, and assumptions.

Reviewer j7Kz initially raised some critical points; however, he did not engage further after the authors offered additional justifications. Overall, the majority of reviewers have acknowledged the paper's merits.